# A long noncoding RNA *HILinc1* enhances pear thermotolerance by stabilizing *PbHILT1* transcripts through complementary base pairing

Yi Zhang[1,2], Shengnan Wang[1,2], Wei Li[1], Shengyuan Wang[1], Li Hao[1], Chaoran Xu[1], Yunfei Yu[1], Ling Xiang[1], Tianzhong Li[1✉] & Feng Jiang[1✉]

As global warming intensifies, heat stress has become a major environmental constraint threatening crop production and quality worldwide. Here, we characterize *Heat-induced long intergenic noncoding RNA 1* (*HILinc1*), a cytoplasm-enriched lincRNA that plays a key role in thermotolerance regulation of pear (*Pyrus* spp.). *HILinc1 Target 1* (*PbHILT1*) which is the target transcript of *HILinc1*, was stabilized via complementary base pairing to upregulate its expression. PbHILT1 could bind to Heat shock transcription factor A1b (PbHSFA1b) to enhance its transcriptional activity, leading to the upregulation of a major downstream transcriptional regulator, *Multiprotein bridging factor 1c* (*PbMBF1c*), during heat response. Transient overexpressing of either *HILinc1* or *PbHILT1* increases thermotolerance in pear, while transient silencing of *HILinc1* or *PbHILT1* makes pear plants more heat sensitive. These findings provide evidences for a new regulatory mechanism by which *HILinc1* facilitates PbHSFA1b activity and enhances pear thermotolerance through stabilizing *PbHILT1* transcripts.

[1] Collage of Horticulture, China Agricultural University, 100193 Beijing, China. [2] These authors contributed equally: Yi Zhang, Shengnan Wang.
✉email: litianzhong1535@163.com; jiangfeng@cau.edu.cn

Suitable temperature is one of the vital environmental conditions for plant growth and development. According to data collected from National Aeronautics and Space Administration, the average global temperature on Earth has increased by around 0.8 °C since 1880[1]. According to the data on Tianqihoubao website (http://www.tianqihoubao.com), the average maximum temperature of major pear-producing areas in China (including Hebei, Anhui, Shandong, Henan, Shanxi and Zhejiang) was 38.8 °C in 2022, which increased about 2.3 °C compared with 2011. The extremely high temperature events are becoming an increasingly challenging abiotic stress that causes great damage to plants including deciduous fruit trees such as pear, by inducing reactive oxygen species (ROS) accumulation, damaging membrane structures, initiating protein misfolding, etc[2,3]. Consequently, plants experience decreased photosynthesis, sunburn, poor pollination and fertilization, and low fruit-setting rates, resulting in a serious decline in agronomic yield and quality[1,2,4–7].

In recent years, large-scale genome-wide studies indicated that thousands of RNAs lacking protein-coding capacity can be transcribed from plant genomes. In particular, long noncoding RNAs (lncRNAs), whose length is >200 nucleotides, have been revealed to play key roles in plants in response to abiotic stress[8,9]. For example, overexpression of *npc536* (*long non-protein coding536*), a natural antisense transcript of *AT1G67930*, resulted in longer root lengths in *Arabidopsis thaliana* under salt stress[10]. Nucleus-located *DROUGHT INDUCED lncRNA* was upregulated by drought, salt, and abscisic acid treatments, promoting *Arabidopsis* tolerance to drought and salt stress[11]. Cold-induced *MADS AFFECTING FLOWERING4 Antisense RNA* (*MAS*) was reported to interact with WD repeat domain 5a (WDR5a), one core component of the COMPASS-like complexes, and positively regulate *MAF4* (*MADS AFFECTING FLOWERING4*) expression by chromatin modification[12]. It has also been reported that some lncRNAs, such as *induced by phosphate starvation 1* (*IPS1*), *cis-NAT$_{AMT1.1}$*, and *TAS3*, take part in nutrient deficiency regulation[13–15]. It is therefore reasonable to explore whether lncRNAs play important roles in heat stress–mediated biological processes. Indeed, there is growing support for a link between lncRNAs and plant thermotolerance. For example, in *Arabidopsis*, *asHSFB2a* (natural antisense transcript of *HSFB2a*) was found to be induced by heat stress and negatively regulate *HSFB2a* expression[16]. *NAT398b/c* (natural antisense transcripts of *MIR398* genes) have been proved to repress miR398b/c biogenesis by impairing the stability of pri-miR398b/c and interfering with its cleavage, thereby attenuating plant thermotolerance[17]. In poplar (*Populus simonii*), TCONS_00202587 functioned as an RNA scaffold to interfere with target gene transcription, and enhanced *Arabidopsis* thermotolerance through overexpression[9]. To date, a number of heat-response lncRNAs have been identified through high-throughput sequencing;[9,16] however, the regulatory mechanism of the lncRNAs in thermotolerance is still largely unknown, especially for long intergenic noncoding RNAs (lincRNAs).

Heat shock transcription factors (HSFs) play important roles in the response and acclimation of eukaryotes under heat stress. Based on their basic structures and evolutionary relationships, HSFs are divided into three classes, HSFA, HSFB, and HSFC, among which the function of HSFA1s play master transcriptional regulators of the heat shock-response (HSR) genes in plants[2,18]. Under room temperature, HEAT SHOCK PROTEIN (HSP)70 and HSP90 bind to HSFA1s to suppress their activities in tomato (*Solanum lycopersicum*) and *Arabidopsis thaliana*[19,20]. As the temperature rising, HSFA1s are released from the inert complex and specifically bind to the heat shock element (HSE) in the promoter region of HSR genes to regulate their expression[2,18,20–27]. HSFA1b directly binds to the promoter of *MBF1c* (*Multiprotein bridging factor 1c*) and stimulates its transcription in *Arabidopsis thaliana*[28]. The plants are survived by a complex regulatory cascade through HSR genes at high temperature by scavenging ROS and repairing cell damage, which underlies the acquisition of thermotolerance[29–31]. Although the majority of HSR genes are modulated by HSFA1s, several HSFs are reported to be involved in the HSR in a HSFA1s-independent manner, such as HSFA4s, HSFA5, and HSFA8[2,20,21,25,32–35]. Whether those HSFA1s-independent HSFs could influence the functions of the HSFA1s under heat stress is yet to be elucidated, however.

Pear is a horticultural crop widely cultivated in the world, and its yield and quality are seriously affected by high temperature. To explore heat resistance mechanism in pear, we conducted transcriptome analysis on 'hongbaoshi' pear under heat stress. Among differentially expressed genes (DEGs), we identified a heat-induced lincRNA, *HILinc1*, in pear (*Pyrus* spp.). *HILinc1* is directly regulated by PbHSFA4b and stabilize *HILinc1 Target 1* (*PbHILT1*) transcripts by complementarily base pairing, leading to the enhancement of its expression level and accumulation of PbHILT1 protein in the nucleus. PbHILT1 functions as a transcriptional assistant to strengthen PbHSFA1b transcriptional activity, resulting in the upregulation of its downstream HSR gene targets, such as *PbMBF1c*, which has a dominant-positive influence on heat tolerance in pear.

## Results

**Identification of heat-induced lincRNA *HILinc1* in pear**. To investigate the influence of high temperature to the pear, 'Conference' (*Pyrus communis*), 'Akizuki' (*Pyrus pyrifolia*), 'Zaojinsu' (*Pyrus* spp.), 'Jinshuisu' (*Pyrus* spp.) and 'Hongbaoshi' (*Pyrus* spp.) were subjected to 38 °C, and all of the five pear cultivars were damaged by heat (Supplementary Fig. 1). After 6 h treatment at 38 °C, the expression of several HSR genes, like *PbMBF1c*, were induced to a high level in 'Hongbaoshi' (Supplementary Fig. 2). To investigate how lncRNAs respond to heat stress in pear, leaves of the crossbreed 'Hongbaoshi' (*Pyrus* spp.) with the strongest heat resistance were collected after 6 h treatment at 38 °C or 25 °C and subjected to high-throughput sequencing. Based on the pipeline (Supplementary Fig. 3a), we found 370 differentially expressed polyadenylated lncRNAs (Supplementary Data 4). Among these, 234 were upregulated (Supplementary Fig. 3b), and were therefore considered as heat-induced lncRNAs, further classified into 137 overlapping, 52 intergenic, 40 natural antisense, and five intronic lncRNAs (Fig. 1a). Among all 52 long intergenic noncoding RNAs, *Linc1* was most abundant under 25 °C (Supplementary Data 4), and was substantially upregulated after the heat treatment (Fig. 1b). In addition, the upregulation of *Linc1* under heat stress could be observed in majority of the pear cultivars (Supplementary Fig. 4a). In conclusion, *Linc1* was a heat-inducible lncRNAs in pear.

To further identify the characteristic of *Linc1*, 5′ and 3′ rapid amplification of cDNA ends (RACE) was used in 'Hongbaoshi' (*Pyrus* spp.), obtaining the full length of 1850 bp (Supplementary Fig. 4b, c). *Linc1* is located on chromosome 5, and its transcript was modified with a poly(A)$^+$ tail and a 5′ 7-methylguanylate cap (Fig. 1c). *Linc1* is unlikely to encode a protein as its coding potential score, calculated via CPC (http://cpc.cbi.pku.edu.cn/)[36], was –1.28 (Fig. 1d), which was under −1, indicating no coding ability. A subcellular distribution analysis showed that *Linc1* was more abundant in the cytosolic fraction than the nuclear fraction (Fig. 1e). We next confirmed the temporal expression pattern of *Linc1* under 38 °C. The results showed that *Linc1* was induced after heat treatment and its expression peaked after 6 h at a level

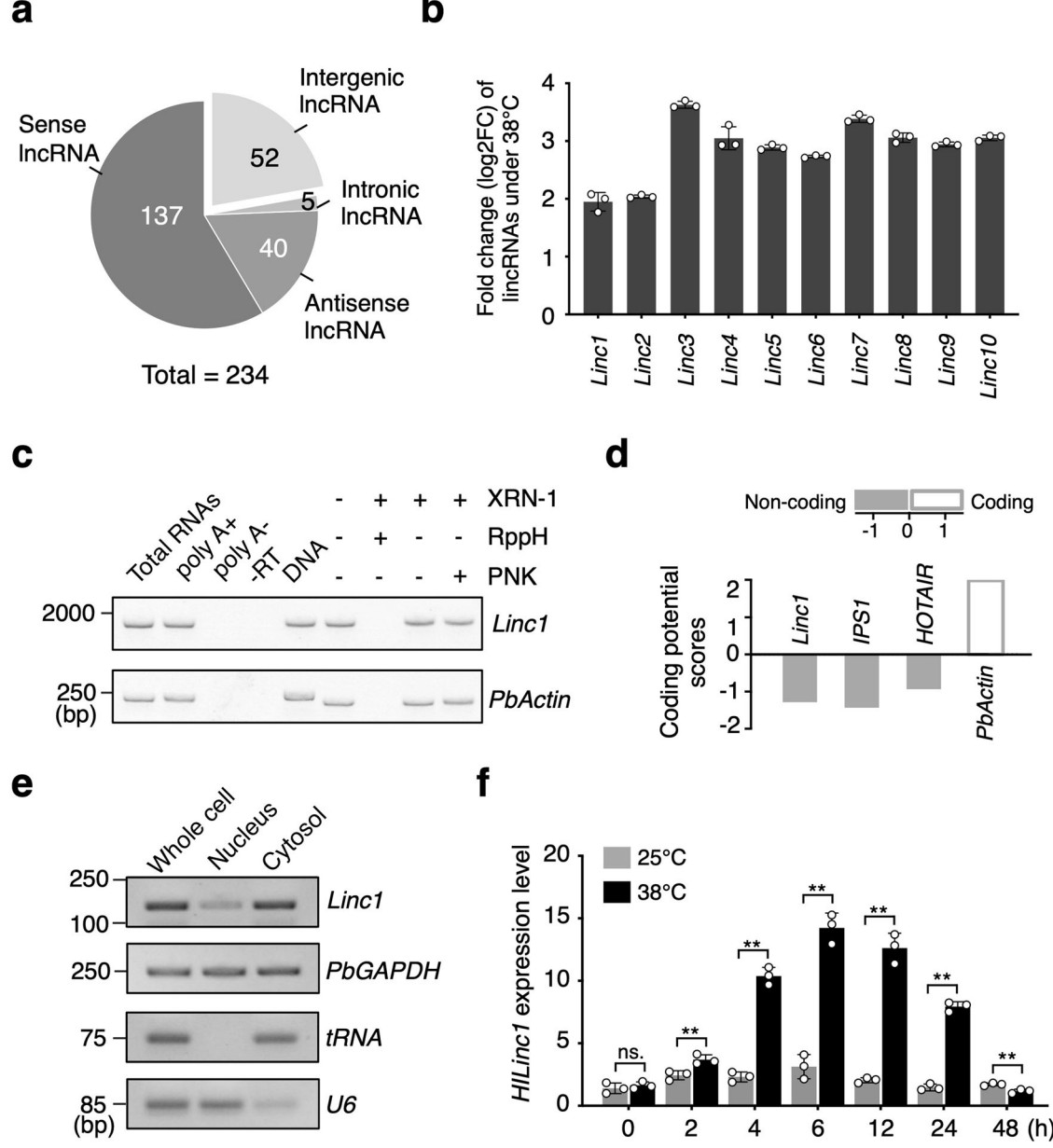

**Fig. 1 Identification of heat-induced HILinc1 in crossbreed 'Hongbaoshi' (Pyrus spp.). a** Classification of upregulated lncRNAs in response to heat treatment. **b** Fold change (log2FC) of the ten most highly expressed lincRNAs after 6 h at 38 °C compared with 25 °C using RT-qPCR. **c** Determination of the 3′ and 5′ end structures of *Linc1*. Random-primed RT-PCR was performed on total RNAs, poly(A)$^+$ RNAs, poly(A)$^-$ RNAs, and RNAs treated (+) or not (−) treated with various enzymes from pear leaves after a heat treatment. XRN-1, 5′–3′ exoribonuclease; RppH, RNA 5′ pyrophosphohydrolase; PNK, T4 polynucleotide kinase. -RT, reverse transcription performed without reverse transcriptase. *PbActin* serves as the control for poly(A)$^+$ and capped RNA. PCRs were performed with DNA of 'Hongbaoshi' for the positive controls. **d** Coding potential of *Linc1*. The CPC program was used for the coding potential score calculation. Transcripts with scores under −1 are classified as noncoding, while those with scores >1 are considered to be coding (Kong et al., 2007). *IPS1* and *HOTAIR* are noncoding representative RNAs and *PbActin* is the coding example. **e** Subcellular distribution of Linc1, as determined using RT-qPCR. *PbGAPDH* is the control for both the nucleus and cytosol distributions. U6 and tRNA are the representatives of the nucleus and cytosol, respectively. **f** Temporal expression pattern of *HILinc1* in 'Hongbaoshi' subjected to 38 °C, as determined using RT-qPCR. The experiments were performed independently three times, and error bars represent the standard deviation. Significant differences were determined using a two-tailed Student's *t* test (*$P < 0.05$, **$P < 0.01$).

about triple that of the control before decreasing (Fig. 1f). Taken together, we identified a lincRNA induced by heat treatment in pear, named as *Heat-induced long intergenic noncoding RNA 1 (HILinc1)*.

**HILinc1 positively regulates pear thermotolerance.** In order to investigate the function of *HILinc1* under heat stress, the expression was regulated using vacuum infiltration via *Agrobacterium tumefaciens* in pear, which were then exposed to 38 °C. Conspicuous differences in heat tolerance were observed between the control (transformed by an empty vector) and transgenic plants (Supplementary Figs. 1 and 5). Leaves of the control plants began wilting at 12 h post heat treatment (HPHT) and started browning at 24 HPHT. By 48 HPHT, the brown area had

expanded and the petioles had wilted, browned, and drooped (Fig. 2a). All *HILinc1*-overexpressing plants exhibited strong heat tolerance and did not undergo leaf blade wilting until 48 HPHT, with no tissue browning. By contrast, the *HILinc1*-silenced plants started wilting (6 HPHT) and browning (12 HPHT) earlier than the control. The browning rate of the *HILinc1*-silenced plants reached 75% accompanied by 25% death by 48 HPHT (Fig. 2b, c).

Furthermore, the soil and plant analyzer development (SPAD) value indicated that the chlorophyll content of *HILinc1*-overexpressing plant leaves was higher than that of the control leaves, while their electrolyte leakage and MDA (malondialdehyde) content declined by 20% and 23%, respectively (Fig. 2d–f). *HILinc1*-silenced plants displayed the opposite changes (Fig. 2d–f).

We next examined several heat-regulated genes to elucidate the influence of *HILinc1* in the heat-response signaling pathway. Overexpressing *HILinc1* resulted in the upregulation of *PbMBF1c*, *PbPIP2A* (*plasmamembrane intrinsic protein 2A*), *PbHSP15.7*, and *PbHSP16.9-I1*, while silencing *HILinc1* suppressed the expression of those four genes (Supplementary Fig. 6). Taken together, these results indicated that *HILinc1* is involved in the fine-tuning of thermotolerance in pear.

**PbHSFA4b is responsible for the transcription of *HILinc1*.** To explore the key transcription factor (TF) controlling *HILinc1* expression under heat stress, PlantTFDB (http://planttfdb.cbi.pku.edu.cn) was firstly employed to analyze the *cis*-acting elements on the promoter sequence of *HILinc1*. A HSE was found in the promoter region from –1080 to –1057 bp upstream of *HILinc1*. In particular, there was also a predicted 342 bp open reading frame containing the HSE domain locating from –1190 to –848 bp upstream of *HILinc1* (Supplementary Fig. 7a). To investigate whether this ORF containing HSE was responsible for *HILinc1* induction under heat stress, *pHILinc1190::GUS* and *pHILinc848::GUS* (*GUS* gene under the control of *HILinc1* promoter with or without the ORF) were constructed and transformed into 'Hongbaoshi' leaves (Fig. 3a). Both histochemical staining and expression analysis showed that the induction of GUS activity was much stronger in leaves expressing *pHILinc1190::GUS* than those expressing *pHILinc848::GUS* after 38 °C treatment (Fig. 3b, c). Furthermore, mutation of the HSE site resulted in a significant decline in GUS activity (Fig. 3b, c), which supported the core role of HSE on the *HILinc1* promoter in heat response.

Among the TF candidates predicted to bind to the HSE on *HILinc1* promoter by PlantTFD, PbHSFA4b showed the highest binding score. To verify the interaction between PbHSFA4b and the *HILinc1* promoter region containing HSE (–1057 to –1080) in vitro, electrophoretic mobility shift assays (EMSAs) were employed. The results showed that PbHSFA4b directly bound to the DNA probe, which was competed by the unlabeled probe (Fig. 3d). And PbHSFA4b failed to bind to the probe containing a mutated HSE site (Fig. 3a, d). Yeast one-hybrid (Y1H) assays presented results consistent with the EMSA (Supplementary Fig. 7b). The result of chromatin immunoprecipitation (ChIP) showed that PbHSFA4b could bind to the promoter of *HILinc1* at 38 °C but not 25 °C in 'Hongbaoshi' (Fig. 3e). In addition, RT-qPCR experiments showed that *PbHSFA4b* was upregulated after a 6 h 38 °C treatment (Supplementary Fig. 7c). Compared with control plants, expression of *HILinc1* was significantly induced in the leaves of *PbHSFA4b*-overexpressing plants under heat stress, while the opposite trend was detected in *PbHSFA4b*-silenced plants (Fig. 3f and Supplementary Fig. 7d). In total, the above data revealed that PbHSFA4b positively regulates the transcription of *HILinc1* in response to heat stress by directly binding to the HSE on its promoter region.

**HILinc1 stabilizes transcripts of its target gene through complementary base pairing.** It has been reported that lincRNAs are able to regulate the expression of neighboring genes[37]. Therefore, 5000 bp both upstream and downstream of *HILinc1* according to pear genome database were scanned for identifying potential targets of *HILinc1* from its neighboring genes. Two ORFs were located upstream and downstream of *HILinc1*, respectively (Fig. 4a). We found no conserved domains in the proteins encoded by the two ORFs, according to CDD (Conserved Domain Database) (https://www.ncbi.nlm.nih.gov/cdd) and Pfam (http://pfam.xfam.org/), and tentatively named them *Pyrus bretschneideri HILinc1 Target 1* (*PbHILT1*) and *HILinc1 Target 2* (*PbHILT2*) (Fig. 4a). In addition to having a similar tissue-specific expression pattern to *HILinc1* in all five tested cultivars of pear (Supplementary Fig. 8), *PbHILT1* was induced in leaves overexpressing *HILinc1* and downregulated when *HILinc1* was silenced (Fig. 4b and Supplementary Fig. 9). By comparison, whether at 25 °C or 38 °C, the expression level of *PbHILT2* was barely influenced by *HILinc1* (Supplementary Fig. 9). Furthermore, *PbHILT1* expression was also increased in *PbHSFA4b*-overexpressing leaves and reduced in *PbHSFA4b*-silenced leaves (Fig. 4c). These results showed that *PbHILT1* was likely regulated by *HILinc1* in responds to heat stress.

Unexpectedly, a fragment of *HILinc1* (from 1348 to 1416 bp) was found to reverse-complement with *PbHILT1* sequence from 21 to 93 bp (Fig. 4d). Northern blot analysis was conducted using the complementary region probes of *HILinc1* and *PbHILT1*, respectively. It was found that the two regions could hybridized into bands of different sizes, indicating that the complementary region did not form double-stranded RNA that could induce RNA degradation (Supplementary Fig. 10). Combined with our previous findings that *HILinc1* positively regulates *PbHILT1*, we raised a hypothesis that *PbHILT1* transcript might be stabilized by *HILinc1* via RNA interaction. To verify this hypothesis, total RNA of 38 °C-treated pear leaves were digested with RNase A/T1 mix and two sets of specific primers respectively against the complementary (set 1) and non-complementary (set 2) sequences were designed for RT-PCR detection. The results showed that the complementary fragment (set 1) survived the degradation by RNase A/T1, whereas the non-complementary fragment (set 2) did not (Fig. 4d). Furthermore, the RNA decay rate of *PbHILT1* was measured in tissue-cultured pear treated with the transcriptional inhibitor actinomycin D. The decline rate of the *PbHILT1* transcripts was slower in *HILinc1*-overexpressing plants than in controls, while by contrast, *HILinc1* silencing caused faster degradation of *PbHILT1* transcripts (Fig. 4e). Deleting the reverse-complement fragment of *HILinc1* destroyed its function in regulating *PbHILT1* (Fig. 4f), which reaffirmed that the regulatory mechanism was likely based on a double-stranded RNA intermediate formed between the *HILinc1* and *PbHILT1* transcripts. Additionally, *PbHILT1* had no effect on *HILinc1* (Supplementary Fig. 11). Taken together, these results suggest that *HILinc1* forms an RNA duplex with *PbHILT1* transcripts through complementary base pairing, which stabilizes *PbHILT1* transcripts.

**PbHILT1 positively regulates pear thermotolerance.** *PbHILT1* expression increased after 38 °C treatment and peaked at 6 HPHT, which was consistent with the expression change of *HILinc1* (Fig. 1f and Supplementary Fig. 12a).

In order to investigate the function of *PbHILT1* under heat stress, *PbHILT1*-overexpressing and -silenced plants were exposed to 38 °C. Obvious differences in heat tolerance were observed between the control and transgenic plants (Fig. 5 and Supplementary Fig. 12c, d). Compared with the control plants, *PbHILT1*-overexpressing plants did not display leaf blade wilting until 48 HPHT, which indicated enhanced heat

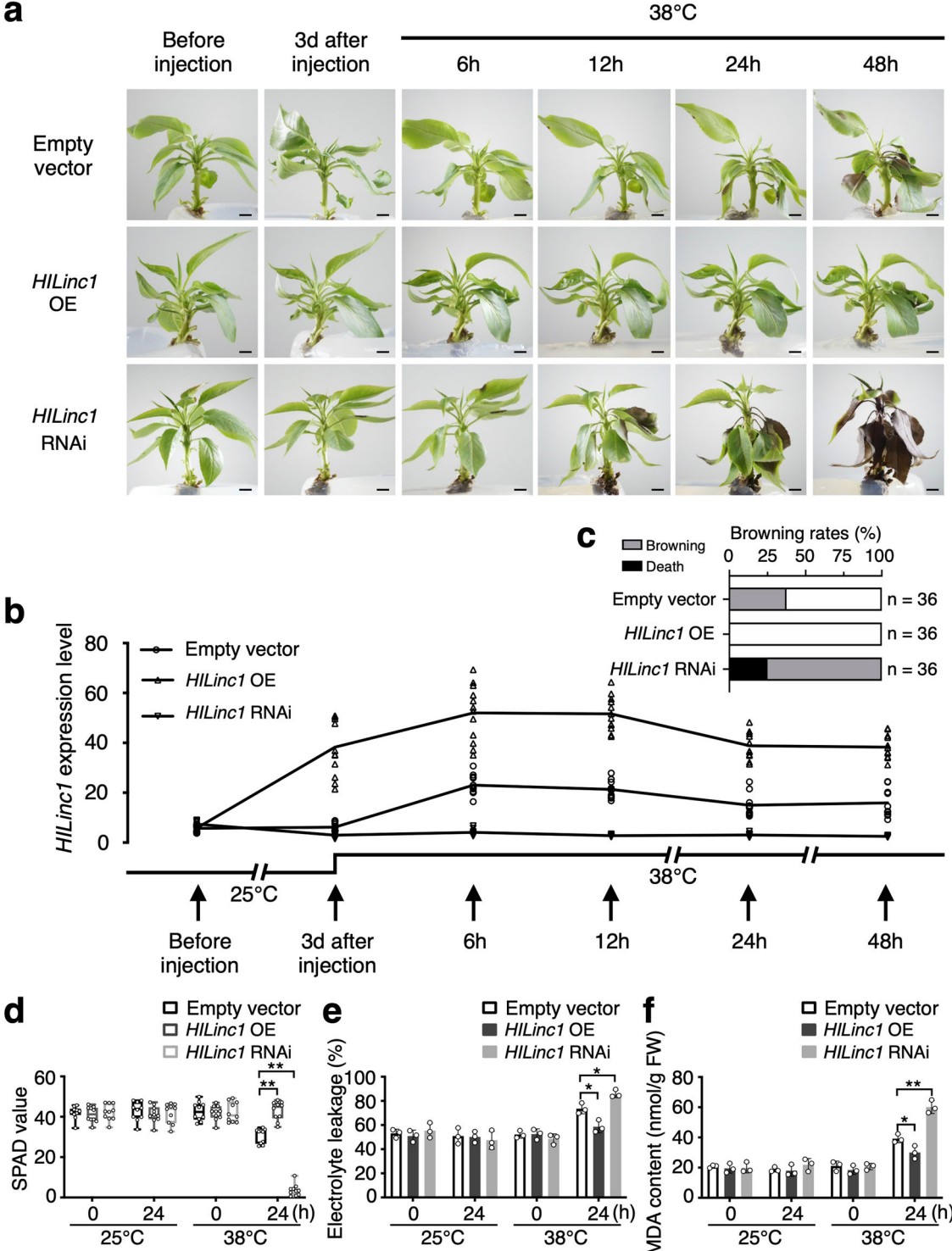

**Fig. 2 Improvement of pear thermotolerance by *HlLinc1*. a–c** Phenotype (**a**), *HlLinc1* expression changes (**b**) and the browning rate (**c**) of *HlLinc1*-overexpressing (OE) and -silenced (RNAi) lines exposed to 38 °C. Lines expressing an empty vector serve as controls. 'Hongbaoshi' pears were vacuum-infiltrated with *p35S::HlLinc1, p35S::RNAi-HlLinc1*, or an empty vector, then subjected to 38 °C or 25 °C (Supplementary Fig. 5) 3 d after the transformation. Leaves before and 3 d after the transformation (0 h post heat treatment), as well as 6, 12, 24, and 48 h post treatment, were harvested for the RT-qPCR analysis. In **a**, bars = 0.5 cm. 36 plants of each genotype were used for the phenotype observation. Representative images are shown. **d–f** SPAD value (**d**), electrolyte leakage (**e**), and MDA content (**f**) of the control, *HlLinc1* OE, and *HlLinc1* RNAi lines after 24 h of 38 °C or 25 °C exposure. Error bars in **b**, **e**, and **f** represent the mean ± SD ($n = 3$), and error bars in **d** indicate the mean ± SD ($n = 10$). Significant differences were determined using a two-tailed Student's *t* test (\**P* < 0.05, \*\**P* < 0.01).

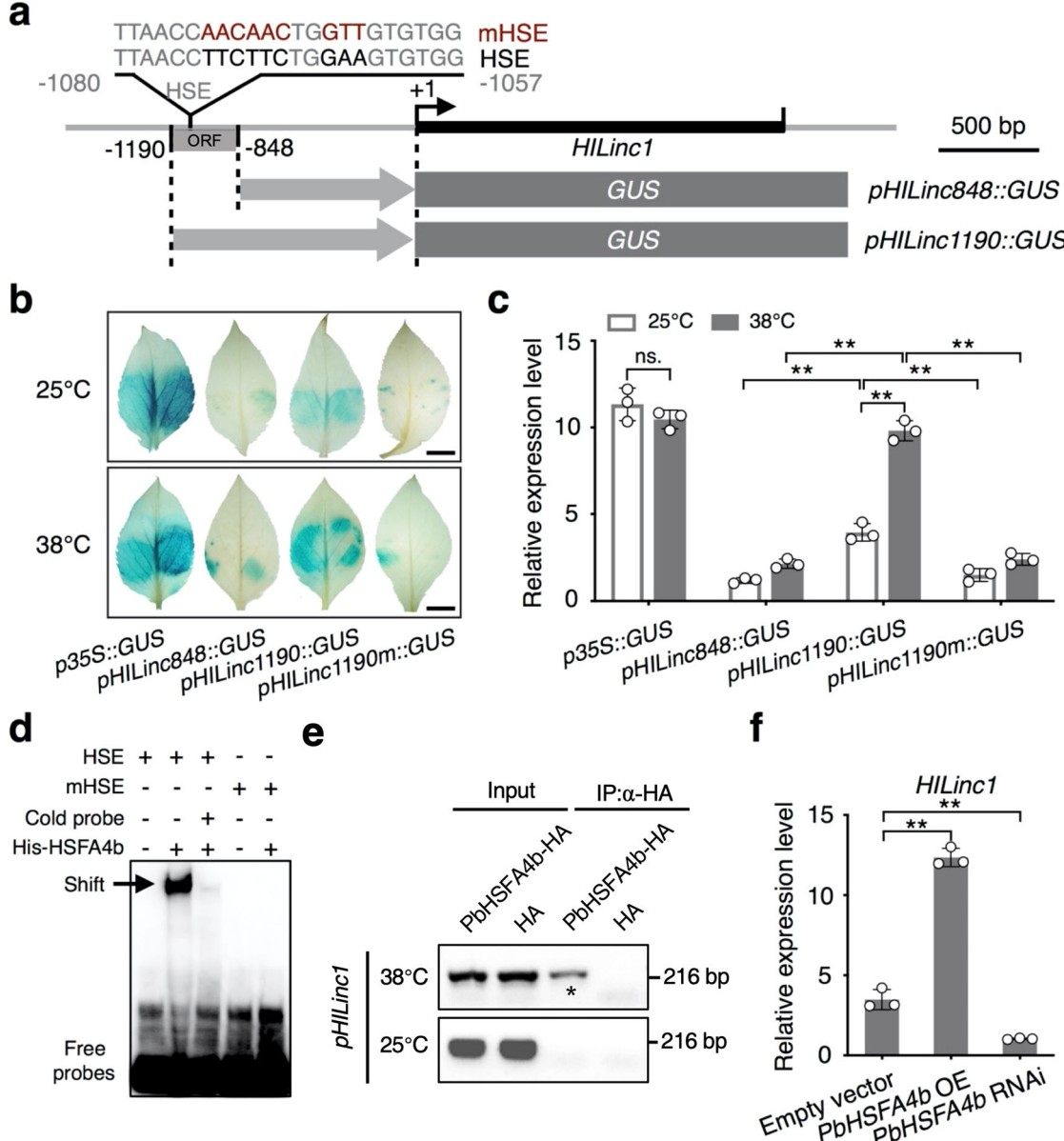

**Fig. 3 PbHSFA4b directly upregulates the expression of *HILinc1*. a** Schematic representation of *pHILinc848::GUS* and *pHILinc1190::GUS* constructs. A 342 bp long open reading frame (ORF) is 1190 bp upstream from the transcription start site of *HILinc1*. The heat stress element (HSE) is contained in the ORF region. The wild-type (lower) and mutant (upper) HSE sequences are highlighted in red. **b**, **c** Histochemical GUS staining (**b**) and *GUS* expression level (**c**) of pear leaves expressing *p35S::GUS*, *pHILinc848::GUS*, *pHILinc1190::GUS*, and *pHILinc1190m::GUS* treated with 38 °C or 25 °C for 6 h. In **b**, bars = 0.5 cm. **d** Electrophoretic mobility shift assay showing the direct binding of PbHSFA4b to the HSE on the *HILinc1* promoter. **e** Binding of PbHSFA4b to the *HILinc1* promoter confirmed by ChIP analysis in 'Hongbaoshi' at 38 °C and 25 °C. **f** Regulation of *HILinc1* by *PbHSFA4b* under a 6 h 38 °C treatment, as determined using RT-qPCR. The experiments were performed independently three times, and error bars represent the standard deviation. Significant differences were determined using a two-tailed Student's *t* test (*$P < 0.05$, **$P < 0.01$).

tolerance. By contrast, *PbHILT1*-silenced plants were observed to be more heat sensitive, which wilted and browned earlier, and suffered great damage at 48 HPHT (Fig. 5A–C). The browning rate of the *PbHILT1*-silenced plants reached 50% accompanied by 25% death by 48 HPHT (Fig. 5D). Correspondingly, leaves from *PbHILT1*-overexpressing plants showed higher SPAD values and lower electrolyte leakage and MDA contents than control, while *PbHILT1*-silenced plants displayed the opposite changes (Fig. 5E–G). These results indicated that *PbHILT1* participates in the regulation of pear thermotolerance.

To confirm whether *PbHILT1* regulates downstream HSR genes, we detected the expression of *PbMBF1c*, *PbPIP2A*, *PbHSP15.7*, and *PbHSP16.9-I1* in *PbHILT1*-overexpressing and -silenced pears. RT-qPCR analysis showed that overexpression of *PbHILT1* led to the upregulation of the four genes, while silencing of *PbHILT1* caused their suppression, which were similar with the effects of *HILinc1* overexpression and silencing, respectively (Fig. 6A and Supplementary Figs. 13–17). Moreover, overexpression of *PbHILT1~mut~* had no influence on *PbMBF1c*, *PbPIP2A*, *PbHSP15.7*, and *PbHSP16.9-I1* (Fig. 6A and Supplementary Fig. 13). Taken together, the results suggest that

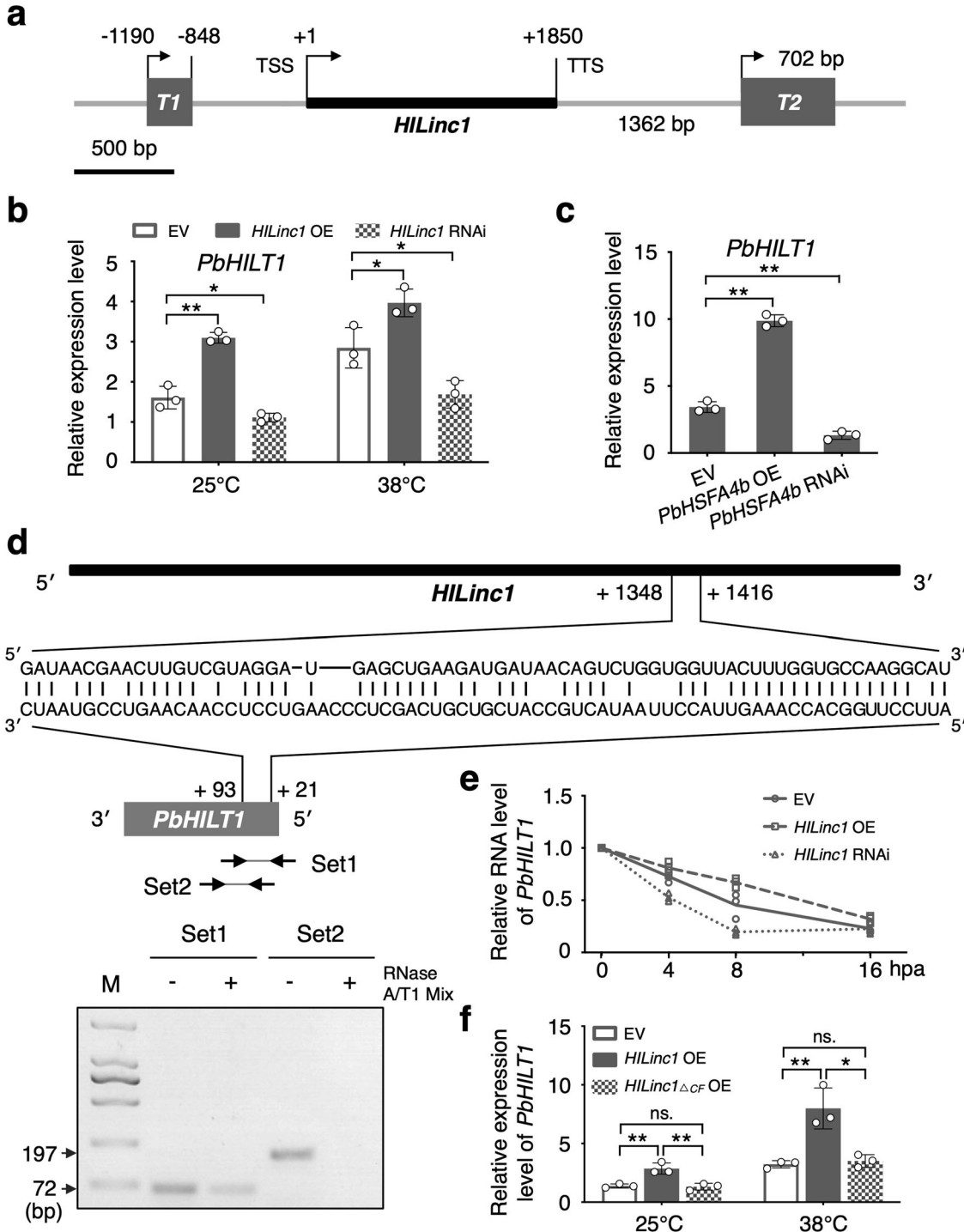

**Fig. 4 Stabilization of the *PbHILT1* transcript by *HlLinc1* through complementary base pairing. a** Schematic position of *HlLinc1* and its two potential target genes in pear. *PbHILT1* is 342 bp long and is located 1190 bp upstream of the transcription start site (TSS) of *HlLinc1*. *PbHILT2* is 702 bp long and 1362 bp downstream of the transcription termination site (TTS) of *HlLinc1*. **b** *PbHILT1* expression in the control, *HlLinc1*-overexpressing (OE), and *HlLinc1*-silenced (RNAi) lines under 25 °C or 38 °C, as determined using RT-qPCR. EV, empty vector (used as control). **c** *PbHILT1* expression in the control, *PbHSFA4b* OE, and *PbHSFA4b* RNAi lines under 25 °C or 38 °C, as determined using RT-qPCR. **d** Schematic diagrams showing the complementary pairing sequences between *HlLinc1* and *PbHILT1* (upper panel) and the detection of RNA duplex transcripts using RT- PCR (lower panel). Set 1 and set 2 are the primers used for the amplification of the complementary and non-complementary fragments of the *PbHILT1* transcript, respectively, after the RNase A/T1 mix treatment. **e** Relative RNA levels of *PbHILT1* in the control, *HlLinc1* OE, and *HlLinc1* RNAi lines after a treatment with actinomycin D (20 µg/mL) for different periods, as determined using RT-qPCR. The data were normalized to the values at 0 h post actinomycin D treatment (hpa). **f** Relative expression levels of *PbHILT1* in the control, *HlLinc1* OE, and *HlLinc1*$_{\triangle CF}$ OE lines under 25 °C or 38 °C, as determined using RT- qPCR. *HlLinc1*$_{\triangle CF}$, *HlLinc1* lacking the complementary fragment. The experiments were performed independently three times, and error bars represent the standard deviation. Significant differences were determined using a two-tailed Student's *t* test (\**P* < 0.05, \*\**P* < 0.01).

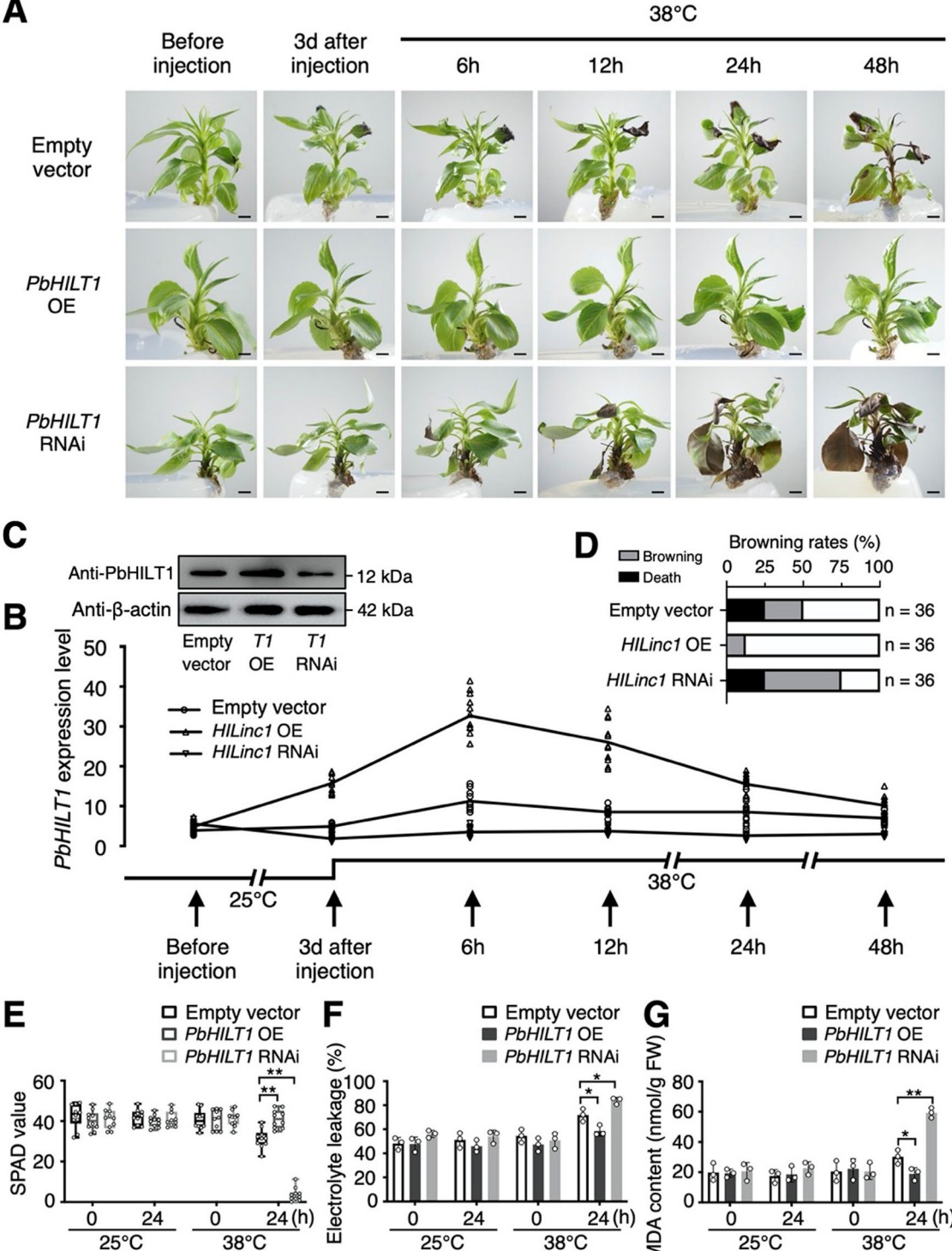

**Fig. 5 Improvement of pear thermotolerance by *PbHILT1*. A–D** Phenotype (**A**), *PbHILT1* expression level (**B**), PbHILT1 protein accumulation (**C**) and the browning rate (**D**) in *PbHILT1*-overexpressing (OE) and -silenced (RNAi) lines exposed to 38 °C. Lines expressing the empty vector serve as controls. 'Hongbaoshi' pears were vacuum- infiltrated with *p35S::PbHILT1, p35S::RNAi-PbHILT1*, or an empty vector, then subjected to 38 °C or 25 °C (Supplementary Fig. 12c) 3 d after the transformation. Leaves before and 3 d after the transformation (0 h post heat treatment), as well as 6, 12, 24, and 48 h post treatment, were harvested for the RT-qPCR analysis. In **A**, bars = 0.5 cm. 36 plants of each genotype were used for the phenotype observation. Representative images are shown. **E–G** SPAD value (**E**), electrolyte leakage (**F**), and MDA content (**G**) of the control, *PbHILT1* OE, and *PbHILT1* RNAi lines after 24 h of 38 °C or 25 °C exposure. Error bars in **B**, **F**, and **G** represent the mean ± SD (*n* = 3), while error bars in **E** indicate the mean ± SD (*n* = 10). Significant differences were determined using a two-tailed Student's *t* test (**P* < 0.05, ***P* < 0.01).

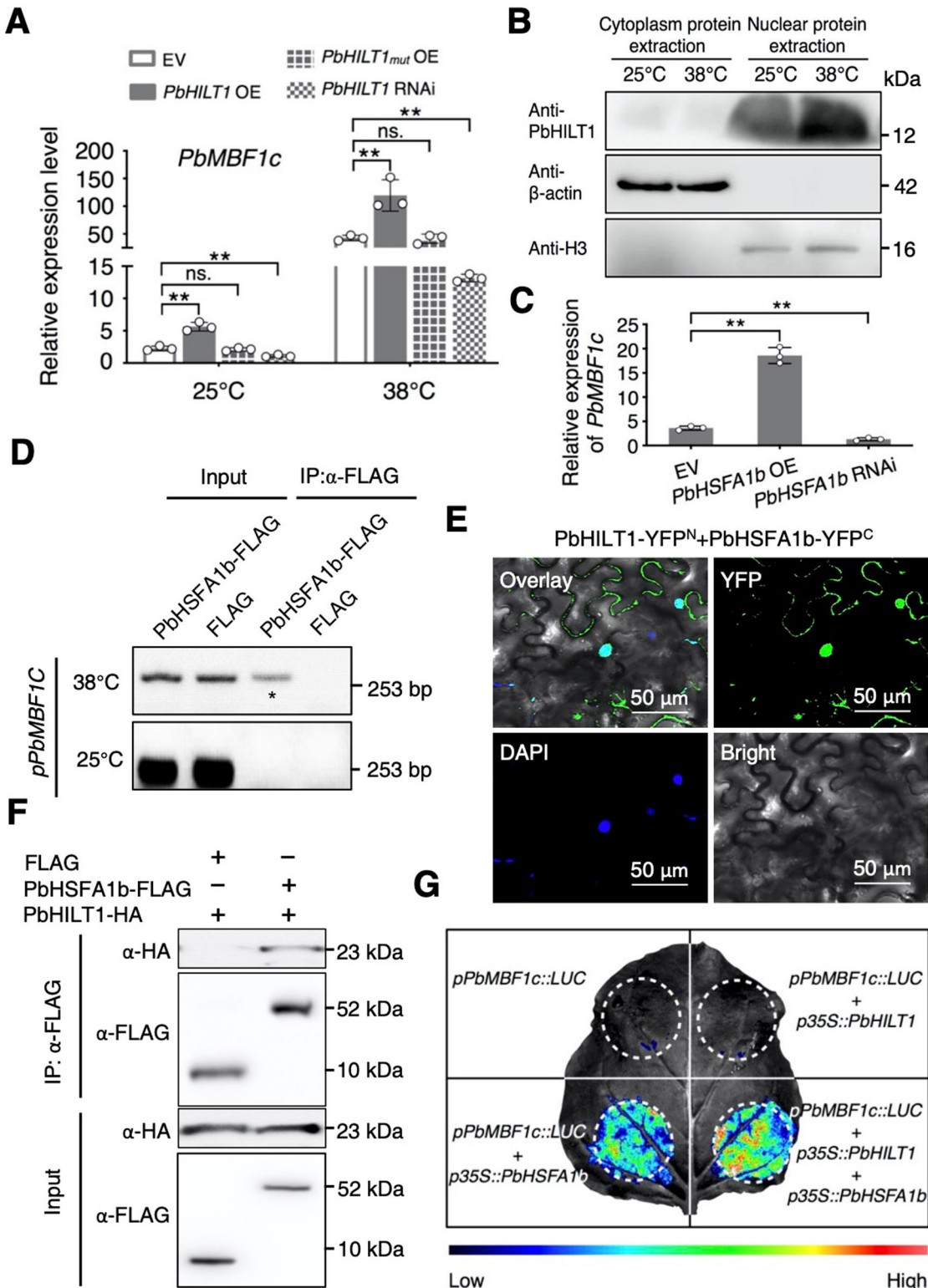

**Fig. 6 PbHILT1 interacts with PbHSFA1b and enhances its transcriptional activity. A** *PbMBF1c* expression in the control, *PbHILT1*-overexpressing (OE), *PbHILT1mut* OE, and *PbHILT1*-silenced (RNAi) lines under 25 °C or 38 °C, detected using RT-qPCR. *PbHILT1mut* was generated using an adenine insertion after the initiation codon of *PbHILT1*. **B** Protein accumulation in nucleus of PbHILT1 after heat stress by western blot. Cytoplasm and nuclear protein was extracted from leaves of 'Hongbaoshi' exposed to 25 °C or 38 °C for 6 h. β-actin and Histone 3 (H3) are the representatives in cytosol and nucleus respectively. **C** *PbMBF1c* expression in the control, *PbHSFA1b* OE, and *PbHSFA1b* RNAi lines subjected to a 6 h treatment at 38 °C, as determined using RT-qPCR. **D** ChIP assays showing the direct binding of PbHSFA1b to the *PbMBF1c* promoter at 38 °C but not 25 °C. **E** Bimolecular fluorescence complementation assays in tobacco leaves with DAPI. Confocal images were captured 48 h after *Agrobacterium* infiltration. Bars = 50 μm. **F** Interaction between PbHILT1 and PbHSFA1b, as determined by Co-IP. **G** Split-luciferase (LUC) assays showing the enhancement of PbHSFA1b transcriptional activity by PbHILT1. *pPbMBF1c::LUC* was co-expressed with *p35S::PbHILT1* and/or *p35S::PbHSFA1b* in tobacco leaves. Bar = 1 cm. The experiments were performed independently three times, and error bars represent the standard deviation. Significant differences were determined using a two-tailed Student's *t* test (*$P < 0.05$, **$P < 0.01$).

*PbHILT1*, the target gene of *HILinc1*, can regulate the expression of *PbMBF1c*, *PbPIP2A*, *PbHSP15.7*, and *PbHSP16.9-I1*.

**PbHILT1 interacts with PbHSFA1b and enhances its transcriptional activity**. Based on our previous findings, several HSR genes were positively regulated by *HILinc1* and its target gene *PbHILT1* (Fig. 6A and Supplementary Figs. 13–17). PbHILT1, despite lacking a nucleus location signal and self-activation activity (Supplementary Fig. 12b), accumulated in the nucleus (Fig. 6B). Accordingly, we hypothesized that PbHILT1 might be carried into the nucleus by a TF to act as a transcriptional assistant. To verify this conjecture, a semi-in vivo pulldown assay was employed for the identification of PbHILT1-associated TFs. Prokaryotic expressed PbHILT1-GST was incubated with total proteins of heat treated 'Hongbaoshi' leaves. According to the mass spectrometry results, PbHSFA1b, which was previously reported to account for the transcription of *PbMBF1c*[28], attracted our attention (Supplementary Data 5). Overexpressing *PbHSFA1b* resulted in increased expression of *PbMBF1c*, while silencing downregulated *PbMBF1c* (Fig. 6C). Chromatin Immunoprecipitation (ChIP) assays presented that PbHSFA1b could directly bind to the promoter region of *PbMBF1c* at 38 °C but not 25 °C in 'Hongbaoshi' (Fig. 6D). There are three HSEs in the promoter region of *PbMBF1c*, and EMSA results showed that PbHSFA1b could directly bind to HSE1 and HSE2 (Supplementary Fig. 18a–c). Additionally, *PbHSFA1b* showed an elevated expression level in response to heat stress (Supplementary Fig. 18d). These results indicate that PbHSFA1b acts as the TF of *PbMBF1c* in pear.

To determine whether PbHILT1 is involved in the regulation of *PbMBF1c* transcription as an assistant with PbHSFA1b, we examined the interaction between PbHILT1 and PbHSFA1. Chromatin Immunoprecipitation (Co-IP), yeast two-hybrid (Y2H), and split-luciferase assays were used to verify the interaction between PbHILT1 and PbHSFA1b, especially in nucleus by Bimolecular fluorescence complementation (BiFC) (Fig. 6E, F, and Supplementary Fig. 19). EMSA assays showed that PbHILT1 had no effect on the binding of PbHSFA1b on *PbMBF1c* promoter (Supplementary Fig. 20). To further inspect the influence of PbHILT1 on PbHSFA1b activity, *pPbMBF1c::LUC* was co-expressed with *PbHILT1* and/or *PbHSFA1b*. The strongest fluorescence intensity was observed when both *PbHILT1* and *PbHSFA1b* were expressed with *pPbMBF1c::LUC* (Fig. 6G), demonstrating that PbHILT1 could enhance the transcriptional activity of PbHSFA1b.

In addition, overexpression of *PbHSFA4b* upregulated *PbMBF1c*, while silencing *PbHSFA4b* resulted in decreased expression of *PbMBF1c* (Supplementary Fig. 21a). However, *PbHSFA1b* expression was not affected in either *PhHSFA4b* overexpressing or silencing line (Supplementary Fig. 21b).

*HILinc1* and *PbHILT1* homologous genes are absent in *Arabidopsis thaliana*. In order to figure out whether *HILinc1* and *PbHILT1* would affect thermotolerance of heterologous plants, we transformed *A. thaliana* with *HILinc1* and *PbHILT1*, and obtained five independent transformants. Compared with wild type, death rates of *HILinc1/PbHILT1* plants were significantly lower after 40 °C treatment for 4 days, followed by recovery under 21 °C for 7 days (Supplementary Fig. 22a, b), demonstrating that these transgenic plants had acquired thermotolerance. Furthermore, expression level of *AtMBF1c* also significantly increased in *HILinc1/PbHILT1* plants (Supplementary Fig. 22c).

Collectively, our data suggest that lincRNA *HILinc1* could promote PbHSFA1b activity and enhance *PbMBF1c* transcription by regulating its target gene, *PbHILT1*, which is beneficial to plant thermotolerance.

## Discussion

In this study, we demonstrated that *HILinc1*, a heat-induced lincRNA in pear, is directly regulated by PbHSFA4b and stabilizes the mRNA of its target gene, *PbHILT1*, through complementary base pairing. PbHILT1 interacts with PbHSFA1b and enhances its transcriptional activity to upregulate *PbMBF1c*, helping to improve thermotolerance in pear (Fig. 7).

It was previously shown that HSFA1b, a member of HSF family, binds to *MBF1c* promoter to increase its expression and activates a series of downstream HSR genes, improving plant thermotolerance[2,18,20,25,28]. HSFA4b is a member of class A HSFs; and the molecular pathway underlying its role in the response to heat stress remains unclear[32]. Based on our data, both *PbHSFA4b* and *PbHSFA1b* respond to heat stress in pear leaves (Supplementary Figs. 7c and 17d), which is consistent with the findings in other species[2,32,38]. Overexpression or silencing *PbHSFA4b* had no effect on *PbHSFA1b* expression (Supplementary Fig. 21b), and the regulation of *PbHSFA1b* expression also did not influence *PbHSFA4b* (Supplementary Fig. 21c), implying that there is no transcriptional regulation existed between *PbHSFA1b* and *PbHSFA4b*. Nonetheless, *PbHSFA4b* could regulate the expression of *PbMBF1c* (Fig. 6A), a direct target of *PbHSFA1b*, for which there are two possible explanations. One is that PbHSFA4b binds to *PbMBF1c* promoter and directly regulates its transcription. The second is that PbHSFA4b takes control of *PbMBF1c* indirectly through its TFs, such as PbHSFA1b. Further studies found that the fluorescence signal was barely observed when *pPbMBF1c::LUC* was co-expressed with *PbHSFA4b* (Supplementary Fig. 21d). It is, therefore, reasonable to speculate that *PbHSFA4b* modulates *PbMBF1c* via another pathway, comprising more regulatory factors, rather than directly activating the transcription of *PbMBF1c*.

In the current study, *HILinc1*, a heat-induced lincRNA in pear, was identified through high-throughput sequencing. The expression of *HILinc1* was directly regulated by PbHSFA4b. *PbHILT1* is located upstream of *HILinc1*, and is the target gene of this lincRNA. Notably, overexpressing or silencing *HILinc1* or *PbHILT1* led to an expression change in *PbMBF1c*. Given the findings above, we conjectured that *PbMBF1c* was regulated by *PbHSFA4b* via the *HILinc1*–*PbHILT1* regulatory module. Although PbHILT1 was shown to accumulate in the nucleus under heat stress, it showed no transcriptional auto-activation capability, which indicated that it was unable to activate the transcription of *PbMBF1c* independently. Further investigation demonstrated that PbHILT1 was able to interact with PbHSFA1b and enhance its transcriptional activity, resulting in the increased expression of *PbMBF1c* (Fig. 7). PbHILT1, first characterized in this study, thus functions to activate PbHSFA1b activity, which is different from the HSF-binding protein (AtHSBP), a negative regulator of HSFA1b previously reported in Arabidopsis[39]. Overexpression of *PbHILT1* improved the thermotolerance of 'Hongbaoshi', while *PbHILT1*-silenced plants showed more serious injury under heat stress. This illustrated that *PbHILT1* plays a dominant role in positively regulating pear thermostolerance. We have also performed analysis in other species, such as apple, tobacco and *Arabidopsis* etc., and found that only apple has homologous *HILinc1*, while lacking of homologous of target gene PbHILT1. Thus, the *HILinc1*-PbHILT1 regulatory pathway is unique in pears. These findings reveal a new heat-response signaling pathway between PbHSFA4b and PbHSFA1b. PbHSFA4b–HILinc1–PbHILT1–PbHSFA1b is likely to be a crucial regulatory module regulating PbHSFA1b and heat tolerance special in pear.

Based on genome location and context, lncRNAs can be classified as overlapping lncRNAs, natural antisense transcripts,

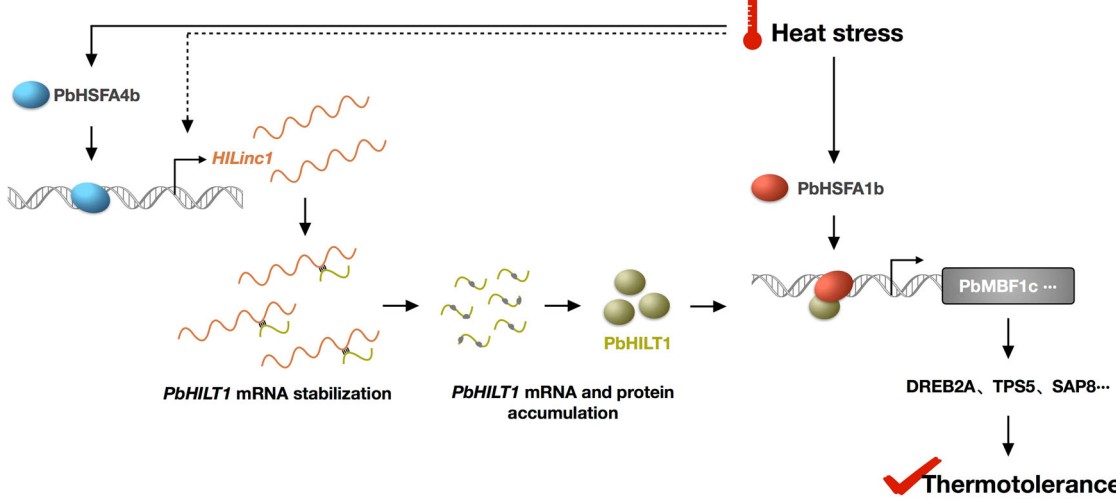

**Fig. 7 A proposed model showing the role of *HILinc1* and *PbHILT1* in the heat response and thermotolerance regulation in pear.** At high temperatures, *HILinc1* is induced by PbHSFA4b and stabilizes *PbHILT1* transcripts by forming an RNA duplex. PbHILT1 interacts with PbHSFA1b and enhances its transcriptional activity, resulting in the upregulation of downstream HSR genes (such as *PbMBF1c*) and the improvement of thermotolerance.

lincRNAs, and intronic noncoding RNAs[12,40]. *HILinc1* belongs to lincRNA. Natural antisense transcripts are the most widely studied lncRNAs in plants, which usually function through regulating their corresponding sense transcripts;[12,17,41–43] however, studies on the regulatory mechanisms of lincRNAs are limited due to the uncertainty of their target genes. In general, there are three approaches to predict the target genes of lincRNAs. First, lincRNAs are likely to regulate neighboring genes[37], so these can be explored as potential targets. The second way is to identify a specific association with the sequences of protein-coding genes, such as the existence of complementary base fragments[44]. The third is to examine correlations in expression patterns between lincRNAs and protein-coding genes[45,46]. To reveal the potential target gene of *HILinc1*, we analyzed its neighboring genes and found two ORFs, one located upstream and one downstream of *HILinc1* (Fig. 4). *PbHILT1* exhibited similar tissue expression specificity to *HILinc1* and was positively regulated by the lincRNA, which was confirmed in several pear cultivars. *HILinc1* contained a fragment that reverse-complemented partial sequences of *PbHILT1*, which was shown to be responsible for forming RNA duplexes with *PbHILT1* transcripts to stabilize them. This type of regulatory mechanism has never been identified in plants; however, similar examples have been reported in animals and microbes, such as *BACE1-AS* and *PTENpg1 asRNA β* in human cells[47,48], and *NfiS* in *Pseudomonas stutzeri*[49,50]. The accumulation of PbHILT1 proteins was observed in *HILinc1*-overexpressing plants, which might be explained by two possibilities. On one hand, *HILinc1* increased *PbHILT1* expression by stabilizing its mRNA, raising the efficiency of ribosome binding and translation. On the other hand, *HILinc1* might not only regulate the transcript level of *PbHILT1*, but also affect the translation efficiency of *PbHILT1* mRNA, similar to the function of *NfiS*;[50] however, this hypothesis requires further exploration. Furthermore, research in mammals showed that nuclear-localized lncRNAs can interact with DNA, RNAs, and proteins to modulate nucleosome incorporation, chromatin structure, and gene transcription, while cytoplasmic lncRNAs are more likely to function in posttranscriptional gene regulation, such as mRNA degradation and translation, or signaling transduction[51]. *HILinc1* was found mainly in the cytoplasm (Fig. 1e), and was shown to participate in heat-responsive signaling pathway by stabilizing the transcripts of its target gene, *PbHILT1* (Fig. 7), which is consistent

with the findings in mammals. In addition, there may be other proteins involved in the formation or unwinding of the RNA duplex between *HILinc1* and *PbHILT1* transcripts, as their bond appears much stronger than general hydrogen bonding. Further investigations are needed to explore the binding and unwinding mechanisms of this special RNA duplex.

Like protein-coding genes, the transcription of lncRNAs is under the control of their promoters. *PbHILT1* was found to be located upstream of *HILinc1*, overlapping with the crucial fragment in the *HILinc1* promoter required for heat responsiveness. PbHSFA4b bound to the HSE contained in the key fragment and enhanced the transcription of *HILinc1* in response to heat stress (Fig. 3). It is very rarely reported in plants that a DNA fragment can be transcribed as a protein-coding gene and simultaneously act as a promoter to control the transcription of downstream genes. A previous study revealed that the expression of the lincRNA *ELENA1* was induced by both elf18 and flg22 in Arabidopsis, with the region containing the CBL6-coding locus in the *ELENA1* promoter being responsible for elf18 and flg22 responsiveness[52], which bore a resemblance to our study.

Pear belongs to perennial woody fruit tree, and its genetic transformation has been reported only a few times in the 'Conforence' variety with low transformation efficiency[53]. In this study, *Agrobacterium tumetobacter* vacuum infiltration method was used to overexpress or silence related genes for functional research, and it was found that the transformation efficiency of this instantaneous transformation method could reach about 80%, which was[54]. However, we also admit that used transformation system was not stable and lasted for a short time. Therefore, in this study, the stable transformation system of *Arabidopsis* was used for further verification, and the same conclusion was obtained as that in pears.

In summary, we identified a heat-responsive lincRNA, *HILinc1*, which was directly regulated by *PbHSFA4b* and could promote PbHSFA1b activity through its target gene, *PbHILT1*; however, there are still some mysteries to be investigated. First, it cannot be excluded that other TFs might also take control of *HILinc1* expression under heat stress. Second, there is a high probability that several regulators may be involved in the formation and unwinding of the RNA duplex between *HILinc1* and *PbHILT1* transcripts. Third, it is unknown whether PbHILT1 can activate TFs other than PbHSFA1b. All these unknown aspects are worth further exploration.

## Methods

**Plant materials and growth conditions**. All pears used in this study, including crossbreeds 'Hongbaoshi' (*Pyrus* spp.), 'Zaojinsu' (*Pyrus* spp.) and 'Jinshuisu' (*Pyrus* spp.), 'Akizuki' (*Pyrus pyrifolia*), and 'Conference' (*Pyrus communis*), were tissue-cultured on Murashige and Skoog (MS) medium containing 6-benzylaminopurine (0.8 mg/L) and 1-naphthylacetic acid (0.1 mg/L) at $24 \pm 1 \,°C$ under long-day conditions (16 h light/8 h dark). The plantlets were transferred to fresh medium every 40 d.

**Heat treatment and thermotolerance assay**. Pear plantlets subcultured for 40 d were transferred to $38 \,°C$ (heat treatment) or $25 \,°C$ (controls). After being treated for different time periods (0, 2, 4, 6, 12, 24, or 48 h), the plant was observed and the leaves were harvested for RNA isolation. Physiological indexes were measured at 24 HPHT. The relative chlorophyll contents of leaves were examined using a SPAD 502 device (Konica Minolta, Osaka, Japan). Electrolyte leakage and MDA contents of the leaves were measured as reported previously[54,55].

After growing at $21 \,°C$ under long-day conditions (16 h light/8 h dark) for 4 weeks, wild type (Columbia) and transgenic *Arabidopsis thaliana* were treated at $38 \,°C$ (heat treatment) or $21 \,°C$ (controls) for 4 d, followed by 7-d-recovery at $21 \,°C$, and the death rates were calculated.

**Sequencing and analysis for the identification of lncRNAs**. An EASYspin RNA Rapid Plant Kit (Biomed Gene Technology, Beijing, China) was used to isolate total RNAs from the leaves of 'Hongbaoshi' at 6 HPHT, and were then treated with DNase I (Biomed Gene Technology, Beijing, China). Samples grown at $25 \,°C$ served as controls. High-purity and high-integrity RNA samples were sent to Gooal Gene Corporation (Wuhan, China) for the RNA library construction and sequencing on an Illumina HiSeq 2500 sequencing platform (Illumina Inc., San Diego, CA, USA). Three biological repeats were performed. The low-quality bases and adapter sequences were discarded from the raw sequencing reads, and the remaining clean reads were mapped to the pear (*Pyrus bretschneideri*) reference genome (http://peargenome.njau.edu.cn/default.asp?d=4&m=2) using STAR version 2.5.3 with default parameters.

The pipeline in Supplementary Fig. 3a was employed to identify heat-responsive lncRNA candidates in pear, based on a previous report[12]. The transcripts with a low abundance ($FPKM \leq 10$), short length (length < 200 nt), or those that overlap with known mRNAs were removed. Moreover, the remaining transcripts were subjected to a coding potential calculation using the Coding Potential Assessment Tool (CPAT, version 1.2.2, http://lilab.research.bcm.edu/cpat)[56] and the coding–noncoding index (CNCI)[57]. Only transcripts with both negative CPAT and CNCI scores were annotated as lncRNAs and used for a further expression analysis.

**5′ and 3′ RACE**. RNA samples were isolated from pear leaves after 6 h treatment at $38 \,°C$. The 5′ RACE was performed with a 5′-Full RACE Kit (Takara Bio, Shiga, Japan) and the 3′ RACE was carried out with a 3′-Full RACE Core Set using PrimeScript™ RTase (Takara Bio). The 5′ and 3′ PCR products were amplified using gene-specific primers (listed in Supplementary Data 1) and cloned into the pMD18-T vector for sequencing.

**RNA isolation and digestion**. Poly(A)$^+$ and Poly(A)$^−$ RNAs were isolated from the total RNAs of heat-treated pear leaves using a polyA Spin™ mRNA Isolation Kit (New England Biolabs, Ipswich, MA, USA). T4 polynucleotide kinase (New England Biolabs), RNA 5′ pyrophosphohydrolase (New England Biolabs), and 5′–3′ exoribonuclease (New England Biolabs) were used for the RNA digestion, according to a previous study[58]. After digestion, the RNAs were purified using the modified cetyltrimethyl ammonium bromide (CTAB) method[59] and subjected to RT-PCR. Primer sequences are provided in Supplementary Data 1.

**Nuclear and cytosolic fractionation**. The fractionation of nuclear and cytosolic components was performed as previously reported[12]. Leaves of pear plantlets were ground to a fine powder after a 6 h heat treatment and mixed with 2 volumes of lysis buffer (250 mM sucrose, 20 mM Tris–HCl [pH 7.4], 20 mM KCl, 2.5 mM MgCl₂, 2 mM EDTA, 5 mM DTT, 25% glycerol, and 40 U/mL RNase inhibitor). A double layer of Miracloth (Merck, Darmstadt, Germany) was used to filter the homogenate. After centrifugation at 13,000 *g* for 10 min at 4 °C, the supernatant was collected as the cytoplasmic fraction. The pellet was washed with nuclear resuspension buffer (20 mM Tris–HCl [pH 7.4], 2.5 mM MgCl₂, 5 mM DTT, 25% glycerol, 2% Triton X-100, and 160 U/mL RNase inhibitor) and resuspended in 500 μL Extraction Buffer II (250 mM sucrose, 10 mM MgCl₂, 10 mM Tris–HCl [pH 8.0], 5 mM β-mercaptoethanol, 1% Triton X-100, 350 U/mL RNase inhibitor, and 1× protease inhibitor) after centrifugation at 1500 *g* for 2 min at 4 °C. The suspension was then overlaid on top of 500 μL Extraction Buffer III (1.7 M sucrose, 2 mM MgCl₂, 10 mM Tris–HCl [pH 8.0], 5 mM β-mercaptoethanol, 0.15% Triton X-100, 350 U/mL RNase inhibitor, and 1× protease inhibitor) and centrifuged at 13,000 *g* for 20 min at 4 °C. The pure nuclear pellet was resuspended in lysis buffer. RNAs in the cytosolic and nuclear fractions were obtained using the modified CTAB method[59] and subjected to RT-PCR analyses. U6 and tRNA were used as nuclear and cytosolic RNA markers, respectively. The primers used in RT-PCR were shown in Supplementary Data 1.

The isolation of nuclear and cytosolic proteins was performed using the Plant Nuclear/Cytosolic Protein Extraction Kit (Bestbio, Shanghai, China), according to the manufacturer's protocol.

**Transient transformation assay**. To evaluate the influences of *HILinc1* and *PbHILT1* in pear thermotolerance, they were cloned into pFGC5941. For their overexpression, the intron region of pFGC5941 was replaced by the full-length sequence of *HILinc1* or *PbHILT1*. For their silencing, the specific fragments of *HILinc1* or *PbHILT1* were cloned into the two flanks of the intron in pFGC5941 in sense and antisense orientations. The empty vector of pFGC5941 was used as the control. *Agrobacterium tumefaciens* cells were transformed with the different constructs. After being cultivated overnight in selection medium, the cells were resuspended in injection buffer (10 mM MgCl₂, 10 mM MES-KOH [pH 5.2], 100 μM acetosyringone). The 40-day-old tissue-cultured pear was completely immersed in the infection solution for infiltrating under a vacuum of 65 kPa for 20 min. The transformed plantlets were cultivated under $25 \,°C$ for 3 d then exposed to $38 \,°C$ for different time periods, with plants continuously grown at $25 \,°C$ used as controls.

**RNase protection assay**. Pear leaves were collected for RNA extraction at 6 HPHT. The RNA was treated with RNase A/T1 mix (Thermo Fisher Scientific, Waltham, MA, USA) at $37 \,°C$ for 30 min, followed by digestion with proteinase K. The RNA was purified using the modified CTAB method[59], and cDNA was synthesized for RT-PCR. The primer sequences are provided in Supplementary Data 1.

**RNA decay assay**. After a 6 h treatment at $38 \,°C$, the plantlets were vacuum-infiltrated for 20 min at 65 kPa in a solution containing 20 μg/mL actinomycin D (Merck). Leaves were harvested before (0 h) and 4, 8, and 16 h after the treatment, and were used for RNA extraction and RT-qPCR assays. The primer sequences are provided in Supplementary Data 1.

**Total RNA extraction and Northern blot analysis**. Total RNA was isolated from 'Hongbaoshi' leaves using a modified cetyltrimeth- ylammonium bromide (CTAB) method[60] and treated with DNase I (Invitrogen) to remove DNA contamination. RNA integrity was verified by electrophoresis on a 1.2% agar gel, and the concentration was measured using an ND-1000 NanoDrop spectrophotometer (Thermo Fisher Scientific). RNA gel blot analysis was performed using a Digoxin Hybridization Detection Kit following the manufacturer's instructions (Mylab; DIGD-120). Approximately 60 μg of RNA was separated in a 15% polyacrylamide gel and electrically transferred to Hybond-N + membranes (GE Healthcare). *HILinc1* and *PbHILT1* probes, including antisense and sense probes, were synthetized with a DIG RNA Labeling Kit (SP6/T7) (Roche) using the primers in Supplementary Data 1.

**Electrophoretic mobility shift assay**. *PbHSFA4b* was cloned into pET-30a to produce the His-PbHSFA4b fusion protein, while *PbHSFA1b* was cloned into pGEX-4T for GST-PbHSFA1b purification. Complementary pairs of 5′ biotin-labeled and unlabeled oligonucleotides (sequences shown in Supplementary Data 1) were annealed in 10× buffer solution (100 mM Tris–HCl [pH 7.5], 10 mM EDTA, and 1 M NaCl) at $75 \,°C$ for 30 min and used as probes. The EMSAs were performed using a LightShift™ chemiluminescent EMSA Kit (Thermo Fisher Scientific). The reaction mixture was mixed with loading buffer and subjected to gel electrophoresis on a 6% polyacrylamide gel at 100 V for 1 h, then transferred to a Hybond-N$^+$ membrane (GE Healthcare, Chicago, IL, USA). After being UV cross-linked, the signal on the membrane was detected according to the manufacturer's protocol.

**Yeast one- and two-hybrid assay**. For the Y1H assays, DNA fragments from the *HILinc1* promoter containing the HSE were amplified and cloned into the pHIS2 vector, serving as the bait construct. For the prey construct, the coding region of *PbHSFA4b* was introduced into the pGADT7 vector. The constructs were transformed into yeast strain Y187 using the LiAc/SSDNA/PEG method[61]. The transformants were grown on synthetic defined (SD)/–Trp–Leu medium and then spotted onto SD/–Trp–Leu–His plates supplemented with 30 mM 3-amino-1,2,4-triazole (3-AT) for high-stringency screening.

For the Y2H assays, *PbHSFA1b* and *PbHILT1* were cloned into pGADT7 and pGBKT7, respectively, and transformed into yeast strain AH109. The transformation method and screening strategy were the same as those used in the Y1H assays.

**GUS staining**. Fragments of different lengths from the *HILinc1* promoter (−1 to −848 or −1190 bp) were cloned into pCAMBIA1305.1 to drive the expression of the β-glucuronidase (GUS) reporter. The *pHILinc848::GUS* and *pHILinc1190::GUS* constructs were transformed into the leaves of 40-day-old 'Hongbaoshi' plantlets via *Agrobacterium*. Three days after infiltration, the plants were exposed to $38 \,°C$ for 6 h. Leaves were collected before and after the treatment for histochemical GUS staining and an expression analysis by RT-qPCR. The GUS staining was performed as previously described[62]. Briefly, the leaves were incubated with X-gluc solution

followed by decoloration using 75% ethanol. The primers used in the RT-qPCR are listed in Supplementary Data 1.

**Split-luciferase assay**. The *PbMBF1c* promoter (1.5 kb upstream of the translation start site) was cloned into pGreenII 0080-LUC to drive the expression of the firefly luciferase reporter. *PbHSFA1b* and *PbHSFA4b* were under the control of the *35 S* promoter in pFGC5941. For the interaction analysis between PbHILT1 and PbHSFA1b, pCAMBIA1300-nLUC and pCAMBIA1300-cLUC were employed. The constructs were transformed into *Agrobacterium* and transiently expressed in *Nicotiana benthamiana* leaves by co-infiltration. Two days later, split-luciferase assays were carried out as previously described[63]. The fluorescence signal was detected on a Tanon 5200 Multi system (Tanon Science and Technology, Shanghai, China).

**Bimolecular fluorescence complementation**. A BiFC was carried out using pCAMBIA1300-YFPn and pCAMBIA1300-YFPc to confirm the interaction between PbHILT1 and PbHSFA1b. Yellow fluorescent protein signals in transformed tobacco leaves were observed using confocal laser microscopy on the Leica TCS SP8 device (Leica Microsystems, Wetzlar, Germany) 2 d after infiltration.

**Semi-in vivo pulldown assay**. *PbHILT1* was cloned into pGEX4T-1, and PbHILT1-GST was purified in a prokaryotic system. Before the elution, the recombinant protein was incubated with total protein extracted from 'Hongbaoshi' leaves at 6 HPHT using a Plant Protein Extraction Kit (Huaxingbio, Beijing, China). The final eluent was collected and sent to the QLBio Corporation (Beijing, China) for mass spectrometry.

**Statistics and reproducibility**. Statistical analyses of data other than transcriptome data were performed with GraphPad Prism 9 software. The number of samples per independent experiment (N) and the specific statistical hypothesis testing method (*t*-test) are described in the legends of the corresponding figures. $P < 0.05$ was considered statistically significant for these comparisons. Data are expressed as mean ± standard deviation (s.d.) values.

**Reporting summary**. Further information on research design is available in the Nature Research Reporting Summary linked to this article.

## Data availability

The RNA-seq data generated in this study are available in the NCBI SRA under accession PRJNA702636. The ID of newly generated plasmids in Addgene were available in Supplementary Data 1. Source data are provided in Supplementary Data 2, 3 and uncropped blots are shown at the end of Supplementary data information. Any other data associated with the findings of this study are available from the corresponding author upon request.

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

## Acknowledgements
We are grateful to Dr. Lijun Wang and Hongliang Zhu for their suggestion to improve the work. This work was supported by the National Key Research and Development Program of China (2018YFD1000103), Natural Science Foundation of China (32272659), the Earmarked Fund for the China Agriculture Research System (CARS-28-08), the 2115 Talent Development Program of China Agricultural University and the Construction of Beijing Science and Technology Innovation and Service Capacity in Top Subjects (CEFF-PXM2019_014207_000032).

## Author contributions
Y.Z. and T.L. conceived the project; Y.Z., S.N.W., and L.H. designed the experiments; Y.Z., S.N.W., and S.Y.W. performed the experiments; C.X., Y.Y, C.Z., and L.X. analyzed the data; Yi Y.Z., F.J., and W.L. wrote and modified the paper.

## Competing interests
The authors declare no competing interests.
