## [Peer Review File · Communications Biology]

Reviewers' comments:

Reviewer #1 (Remarks to the Author):

The authors of this manuscript have done a range of experiments to confirm a model where a lncRNA from pear enhance pear thermotolerance.

First, on the technical side - the manuscript DO has to undergo English proofreading because while some parts are written nicely some other parts are littered with English mistakes making it sometimes difficult to follow.

The manuscript is extensive in terms of the list of experiments performed. In general, in my view this is a nice piece of work, below I have a list of points I would like to be addressed by the authors:

Fig1 A

the authors present RT-qPCR data on 10 lncRNA.

Please provide detailed coordinates and identification data for the lncRNA so others can also use them...

2. the way the data is presented suggests that the expression of different lncRNA can be compared. But as different primers were used unless the author does a genomic DNA normalization, they can compare the induction between the different lncRNA but NOT the absolute or relative level of expression between the different lncRNA.

Fig2a relays on transient transformation. As I am not familiar with the pear system I would like the authors to amend the text by providing clear ref for the method and discussing the method in the context of transformation efficiency etc.

Please provide quantification of the heat resistance phenotype - one picture of a SINGLE plant is not enough...

Fig 3a the description in the main manuscript text of this figure in respect to the T1 orf described on the next figure is not clear. I suggest deleting the mentioning of the upstream orf in the description of this data in the text and only discussing it with the next figure?

Fig 4

panel d. Is the amplicon's position on the schematic correct with respect to each other? In other words, do they overlap? if they do overlap so much, then why is the signal only positive for Set1?

e. the RNA stability data is usually shown on a log2 scale allowing the reader to assess the linearity of the measurement

In general, While I liked the breadth of the study, I conclude that:

1. the authors should limit their claims by acknowledging that a transient system is less trust-able than stable mutants - clearly not available for pear in a reasonable time frame.

2. Some of the key experiments need replicates to justify the conclusions - as described above

3. Fig.S6 the tile plot does not contain information about the error, consider changing the plot type or at least provide information about the result of a statistical test, sample size, and gene used for normalization.

Reviewer #2 (Remarks to the Author):

In the study, the authors report a heat-induced lncRNA (HILinc1) that can improve thermotolerance of pear (*Pyrus* spp.). They found that the enhancement of heat tolerance was attributed to its target gene PbHILT1 interacting with PbHSFA1b, and thereby enhancing its transcriptional activity. Overall, this is a well performed study with convincing work that provides new proofs for the lncRNA regulatory networks in plant heat response and tolerance. Given the pervasive lncRNAs involved in the stress response, I think the findings should be of interest to community of the heat response in plants.

The authors need to consider the following questions.

HILinc1 was screened out from leaves of the crossbreed 'Hongbaoshi'. Did the authors investigate the HILinc1 expression in other four pear cultivars under 38°C? If HILinc1 also showed highly expression in all pear cultivars under heat, the results and findings could be applicable for the pear species. Otherwise, the HILinc1 results maybe specific to the heat-resistant cultivar 'Hongbaoshi'.

Plant materials : in Figure 2 and Figure 5, where did the tissue-cultured plantlets come from ? Callus or explant buds? How to ensure the consistent status of these plantlets?

Since the pear (*Pyrus* spp.) is not a model plant species, the authors should search the obtained lncRNA (HILinc1) for homologous sequence in the plant lncRNA database. Although the lncRNA sequence conservation maybe low, if HILinc1 has homologous sequence in other species, it is interesting to know the HILinc1 regulatory is a conservative mechanism in plants, or only species specific to the pear.

The authors transformed *Arabidopsis thaliana* with HILinc1 and PbHILT1, and found HILinc1 and PbHILT1 transformation enhanced *Arabidopsis* thermotolerance. In Supplementary Figure 21 A, the WT, OE1 and OE2 seedlings should be grown in a common pot for heat treatment, rather than growing them separately for heat treatment.

The reason why the pear species were chosen for this study should be given in the Introduction.

Reviewer #3 (Remarks to the Author):

The paper entitled "A long noncoding RNA HILinc1 enhances pear thermotolerance by stabilizing PbHILT1 transcripts through complementary base pairing" describes very interesting interaction between long noncoding RNA which is induced by heat stress and its partner mRNA encoding PbHILT1. This discovery shows how "many novel" function can perform RNA. In my opinion the paper can be published in *Communications Biology* but only after revision and clarification of some observation. The paper is well written and experiments were well planned and performed especially experiment showing the promoter activity of HILinc1 with or without orf sequence(GUS activity).

The most important questions which should be answered are:

1. How is it possible that interaction between HILinc1 and PbHILT1 does not induce siRNA production and further down-regulation expression of both RNA. A. Did you analyze small RNA pool to show that there are no siRNA created from dsRNA, B. Are there any small RNAs created from interacting ssRNAs, C. Could you show alignment between both sequences, on Figure 4 d there is only partial alignment? D. Please also explain what does it mean thermotolerance for pear i.e. particular pear cultivars with higher thermotolerance produce more fruits under high temperature, give more fruits in warmer part

of the country?

Moreover there are some notes which should be explained in the revised manuscript:

Line 31 - ...around 0.8°C since 1880 (Janni et al., 2020)...- such difference between 1880 and 2020 could be explained by temperature measurement method.

Line 91 - ...HSFA4s, HSFA5, and HSFA8(Busch...- lack of space.

Line 96 -In the current study, we identified a heat-induced lincRNA, HILinc1, in pear... - please explain in the text why did you study pear, whether this species is especially sensitive to high temperature.

Line 114 - ...heat resistance ...- I do not understand. Do you think that this cultivar is not affected by heat or is very sensitive to heat i.e. HILinc1 is in this cultivar highly up- or down-regulated compared to the other cultivars studied.

Line 127 - ...To further identify the characteristic of HILinc1... - does this transcripts have any splice variants. Could you show similar sequences (alignment) derived from the other species.

Line 201 - ...neighboring genes(Guttman and Rinn, 2012)... - lack of space.

Line 438 - ... what is the age/stage of the analyzed pear plants according to the pear developmental stages.

Line 439 - ...25°C (controls)... - what is the normal temperature for the pear growth?

Line 481 - ...After digestion, the RNAs... - what was the RNA quality measured using Agilent Bioanalyzer, RIN?

Reviewer #1 (Remarks to the Author):

The authors of this manuscript have done a range of experiments to confirm a model where a lincRNA from pear enhance pear thermotolerance.

First, on the technical side - the manuscript DO has to undergo English proofreading because while some parts are written nicely some other parts are littered with English mistakes making it sometimes difficult to follow.

Answer: We are very sorry for the trouble caused by our language. The language in our manuscript has been modified by professionals.

The manuscript is extensive in terms of the list of experiments performed. In general, in my view this is a nice piece of work, below I have a list of points I would like to be addressed by the authors:

Fig1 A

the authors present RT-qPCR data on 10 lincRNA.

Please provide detailed coordinates and identification data for the lincRNA so others can also use them...

Answer: The detailed coordinates and identification data for the lincRNAs were shown in Supplementary Table1. We have explained this in the text shown in red at line 139.

2. the way the data is presented suggests that the expression of different lincRNA can be compared. But as different primers were used unless the author does a genomic DNA normalization, they can compare the induction between the different lincRNA but NOT the absolute or relative level of expression between the different lincRNA.

Answer: Thanks for your suggestion. We have changed the presentation format of the data to compare the fold change of different lincRNAs under 38°C (Fig. 1B).

Fig2a relies on transient transformation. As I am not familiar with the pear system I would like the authors to amend the text by providing clear ref for the method and discussing the method in the context of transformation efficiency etc.

Answer: We added some details in the material method and cited references. In the transient transformation of pears, each pear tissue culture seedling

infected by *Agrobacterium tumefaciens* was subjected to heat treatment after quantitative analysis of gene expression, and the transformation efficiency could reach more than 80%. We have added these discussions in the text at line 437-446 shown in red.

Please provide quantification of the heat resistance phenotype - one picture of a SINGLE plant is not enough...

Answer: We have added the quantification of the heat resistance phenotype in Fig. 2C and 5D.

Fig 3a the description in the main manuscript text of this figure in respect to the T1 orf described on the next figure is not clear. I suggest deleting the mentioning of the upstream orf in the description of this data in the text and only discussing it with the next figure?

Answer: Thanks for your suggestion. While the location of the T1 ORF cannot be deleted in the description of Fig. 3A, because the segmentation verification of the promoter is based on the location of this ORF. At line 188 - 195, we have added a description of this ORF to make it more explicit.

Fig 4

panel d. Is the amplicon's position on the schematic correct with respect to each other? In other words, do they overlap? if they do overlap so much, then why is the signal only positive for Set1?

Answer: Set1 and set 2 were in the correct position on the pattern diagram, they overlapped. The primers used to amplify set 1 were in the complementary region in *PbHILT1*, while the forward primer used to amplify set 2 was in the non-complementary region, which was the single-stranded RNA region and could be digested by RNase A/T1 mix, so the fragment of set 2 could not be amplified.

e. the RNA stability data is usually shown on a log2 scale allowing the reader to assess the linearity of the measurement

Answer: Thanks for your suggestion. The presentation format of the RNA stability data referred to the paper 'Global identification of *Arabidopsis* lncRNAs reveals the regulation of *MAF4* by a natural antisense RNA', which was also be listed in our references in this manuscript.

In general, While I liked the breadth of the study, I conclude that:

1. the authors should limit their claims by acknowledging that a transient system is less trust-able than stable mutants - clearly not available for pear in a reasonable time frame.

Answer: Thank you for your comments. Since it is not possible to obtain mutants from pears, we verified our conclusion by the stable transformation system of *Arabidopsis thaliana* at the end of the paper in Supplementary Fig. 22. Of course, we are very grateful for your suggestions. We also analyzed the results of transient transformation of pears in the discussion to limit our claims, in red font, at line 442-446.

2. Some of the key experiments need replicates to justify the conclusions - as described above

Answer: We have replicated the key experiments related to *HILinc1* and PbHILT1 in this research, and the results obtained are consistent with the previous. These data have been supplemented in Fig. 2 and 5. In total, 12 replications were performed for each experiment and 3 plants were contained in each replication.

3. Fig.S6 the tile plot does not contain information about the error, consider changing the plot type or at least provide information about the result of a statistical test, sample size, and gene used for normalization.

Answer: Thanks for your suggestion. We have changed the presentation format of the data (Supplementary Fig. 6).

Reviewer #2 (Remarks to the Author):

In the study, the authors report a heat-induced lncRNA (HILinc1) that can improve thermotolerance of pear (*Pyrus* spp.). They found that the enhancement of heat tolerance was attributed to its target gene PbHILT1 interacting with PbHSFA1b, and thereby enhancing its transcriptional activity. Overall, this is a well performed study with convincing work that provides new proofs for the lncRNA regulatory networks in plant heat response and tolerance. Given the pervasive lncRNAs involved in the stress response, I think the findings should be of interest to community of the heat response in plants.

The authors need to consider the following questions.

HILinc1 was screened out from leaves of the crossbreed 'Hongbaoshi'. Did the authors investigate the HILinc1 expression in other four pear cultivars under 38°C? If HILinc1 also showed highly expression in all pear cultivars under heat, the results and findings could be applicable for the pear species. Otherwise, the HILinc1 results maybe specific to the heat-resistant cultivar 'Hongbaoshi'.

Answer: Thanks for your suggestion. We have examined the expression pattern of *HILinc1* at high temperature in many pear varieties and found that the upregulation of *HILinc1* under heat stress could be observed in majority of the pear cultivars detected. The results were shown in Supplementary Fig. 4A, which has been updated. Pear is a fruit tree with complex genetic background, and the heat response pathway is regulated by many factors. The expression of *HILinc1* may not only be affected by temperature, but also regulated by many other factors. As a result, *HILinc1* may exhibit different expression trend in different pear varieties.

Plant materials: in Figure 2 and Figure 5, where did the tissue-cultured plantlets come from? Callus or explant buds? How to ensure the consistent status of these plantlets?

Answer: In our study, the selected pear tissue-cultured plantlets were obtained from explant buds through propagation culture. Tissue-cultured plantlets used in each experiment were subcultured for 40 days, with similar growth conditions, i.e. stem diameter, seedling height and number of leaves for further heat treated. Since the pear (*Pyrus* spp.) is not a model plant species, the authors should search the obtained lncRNA (HILinc1) for homologous sequence in the plant lncRNA database. Although the lncRNA sequence conservation maybe low, if HILinc1 has homologous sequence in other species, it is interesting to know the HILinc1 regulatory is a conservative mechanism in plants, or only species specific to the pear.

Answer: We have analyzed the sequence of *HILinc1* using Nucleotide BLAST software of NCBI, and only one homologous lncRNA was found in apple (*Malus × domestica* Borkh.). However, there was no homologous sequence of the target gene *PbHILT1* in apple. Therefore, HILinc1-HILT1 regulatory module participating in heat response is unique in pears. We have also discussed this part at line 377-384.

The authors transformed *Arabidopsis thaliana* with HILinc1 and PbHILT1, and found HILinc1 and PbHILT1 transformation enhanced *Arabidopsis* thermotolerance. In Supplementary Figure 21 A, the WT, OE1 and OE2 seedlings should be grown in a common pot for heat treatment, rather than growing them separately for heat treatment.

Answer: We have repeated the experiments in which WT and *HILinc1+PbHILT1* overexpressed lines were grown in the same pot for heat treatment as shown in Supplementary Figure 22A.

The reason why the pear species were chosen for this study should be given in the Introduction.

Answer: Pear is a horticultural crop widely cultivated in the world, and its yield and quality are seriously affected by high temperature. In production, we found that 'hongbaoshi' pear was more heat-resistant compared with other varieties. To explore its heat resistance mechanism, we conducted transcriptome analysis on 'hongbaoshi' pear under heat stress. And these descriptions have been added to the introduction as shown in red at line 116.

Reviewer #3 (Remarks to the Author):

The paper entitled “A long noncoding RNA HILinc1 enhances pear thermotolerance by stabilizing PbHILT1 transcripts through complementary base pairing” describes very interesting interaction between long noncoding RNA which is induced by heat stress and its partner mRNA encoding PbHILT1. This discovery shows how “many novel” function can perform RNA. In my opinion the paper can be published in Communications Biology but only after revision and clarification of some observation. The paper is well written and experiments were well planned and performed especially experiment showing the promoter activity of HILinc1 with or without orf sequence(GUS activity).

The most important questions which should be answered are:

1. How is it possible that interaction between HILinc1 and PbHILT1 does not induce siRNA production and further down-regulation expression of both RNA.

A. Did you analyze small RNA pool to show that there are no siRNA created from dsRNA,

B. Are there any small RNAs created from interacting ssRNAs,

Answer: Thanks for your questions, it's a good suggestion. We have designed probes for the complementary region and conducted northern-blot analysis. The results showed that the complementary region was not cleaved to form siRNAs (Supplementary Fig. 10). In that case, the interaction between *HILinc1* and *PbHILT1* does not induce siRNA production and further down-regulation expression of both RNAs. Instead, as shown in figure 4, the transcripts of *PbHILT1* were stabilized by HILinc1 through complementary base pairing. And the similar mechanism has also been identified in animals and microbes as discussed in line 238-242.

C. Could you show alignment between both sequences, on Figure 4 d there is only partial alignment?

Answer: On Fig. 4D, the length of thick black line and grey rectangle represented the length of *HILinc1* and *PbHILT1* respectively. The fragment of *HILinc1* (from 1348 to 1416 bp) was the only region found to reverse-

complement with *PbHILT1*. The sequence shown in Fig. 4D. was the full alignment of complementary region between *HILinc1* and *PbHILT1*.

D. Please also explain what does it mean thermotolerance for pear i.e. particular pear cultivars with higher thermotolerance produce more fruits under high temperature, give more fruits in warmer part of the country?

Answer: In this manuscript, pear seedlings with higher thermotolerance could maintain normal growth conditions under high temperature. Those seedlings were less damaged by heat stress and were able to produce more fruits under high temperature compared with heat-susceptible ones.

Moreover there are some notes which should be explained in the revised manuscript:

Line 31 - ...around 0.8°C since 1880 (Janni et al., 2020)...- such difference between 1880 and 2020 could be explained by temperature measurement method.

Answer: This is the reference we looked up about global temperature change. Sorry, we have no idea about what your question means.

Line 91 - ...HSFA4s, HSFA5, and HSFA8(Busch...- lack of space.

Answer: Thanks for your suggestion. We have checked the full text and corrected such mistakes.

Line 96 -In the current study, we identified a heat-induced lincRNA, *HILinc1*, in pear... - please explain in the text why did you study pear, whether this species is especially sensitive to high temperature.

Answer: Pear is a horticultural crop widely cultivated in the world, and its yield and quality are seriously affected by high temperature. In production, 'hongbaoshi' pear was found to be more heat-resistant compared with other varieties. To explore its heat resistance mechanism, we conducted transcriptome analysis on 'Hongbaoshi' pear under heat stress. And these descriptions have been added to the introduction in red at line 116.

Line 114 - ...heat resistance ...- I do not understand. Do you think that this cultivar is not affected by heat or is very sensitive to heat i.e. *HILinc1* is in this cultivar highly up- or down-regulated compared to the other cultivars studied.

Answer: ‘Hongbaoshi’ is more heat-resistant means that, it was less damaged, which was reflected by less brown leaves, under high temperature compared with other varieties (Supplementary Fig. 1). In ‘Hongbaoshi’, *HILinc1* was the most abundant lincRNA under 25°C (Supplementary Table 1), and was substantially upregulated after the heat treatment (Fig. 1B), regulating downstream genes and preventing plants from being damaged by heat (Fig. 7). In addition, the upregulation of *HILinc1* under heat stress could be observed in majority of the pear cultivars detected in field (Supplementary Fig. 4A). Pear is a fruit tree with complex genetic background, and the heat response pathway is regulated by many factors. The expression of *HILinc1* may not only be affected by temperature, but also regulated by many other factors. As a result, *HILinc1* may exhibit different expression trend in different pear varieties.

Line 127 - ...To further identify the characteristic of HILinc1... - does this transcripts have any splice variants. Could you show similar sequences (alignment) derived from the other species.

Answer: HILinc1 did not have any splice variants and we have analyzed the sequence of *HILinc1* using Nucleotide BLAST software of NCBI, and only one homologous lincRNA was found in apple (*Malus × domestica* Borkh.). However, there was no homologous sequence of the target gene *PbHILT1* in apple. Therefore, HILinc1-HILT1 regulatory module participating in heat response is unique in pears. We have also discussed this part at line 377-384.

Line 201 - ...neighboring genes(Guttman and Rinn, 2012).... – lack of space.

Answer: Thanks for your suggestion. We have checked the full text and corrected such mistakes.

Line 438 - ... what is the age/stage of the analyzed pear plants according to the pear developmental stages.

Answer: Pear growth process includes seedling, growth, flowering, fruit-dropping, fruit expansion, fruit maturity. The material used in our study is tissue-cultured plantlets from explants buds of annual branches, which are in the growth stage, i.e. the vegetative growth stage. Tissue-cultured plantlets used in each experiment were subcultured for 40 days, with similar growth conditions, i.e. stem diameter, seedling height and number of leaves for further heat treated.

Line 439 - ...25°C (controls)... - what is the normal temperature for the pear growth?

Answer: The normal temperature for the pear growth is 25°C, especially for tissue-cultured plantlets.

Line 481 - ...After digestion, the RNAs... - what was the RNA quality measured using Agilent Bioanalyzer, RIN?

Answer: Sorry, our research group doesn't have Agilent Bioanalyzer, RIN, so we didn't do the measurement. However, NanoDrop was used to test the RNA quality and the result showed that, after digestion, the concentration of RNAs dropped while the quality still reached accepted level.

Reviewers' comments:

Reviewer #1 (Remarks to the Author):

I appreciate the corrections made by the authors. the additional explanations and data provided. I also asked the author to limit their claims about the HILinc1. The authors have now provided a section in the discussion about the limitation of their methodology. I however would recommend them too also include the reference to the "transient nature of the method" in the abstract:

I would like to suggest amending the following sentence:

"Overexpressing either HILinc1 or PbHILT1 increases thermotolerance in pear, while silencing HILinc1 or PbHILT1 makes pear more heat sensitive. "

to

"Transient overexpression of either HILinc1 or PbHILT1 increases thermotolerance in pear, while transient silencing of HILinc1 or PbHILT1 makes pear plants more heat sensitive. "

Please also correct the sentence:

line 442 "However, we also admitted that this transformation system was not stable and lasted for a short time with stable functions."

to

line 442 "However, we also admit that used transformation system was not stable and lasted for a short time."

Reviewer #3 (Remarks to the Author):

Well done. I accept the revised article but still I have four notes, see below

Page 2, Line 40 - ... which was the target gene – rather target transcript.

Page 2 Line 54 - ...around 0.8°C since 1880 (Janni et al., 2020)...- such difference between 1880 and 2020 could be explained by temperature measurement method. Answer: This is the reference we looked up about global temperature change. Sorry, we have no idea about what your question means.

Comment: You show that from 1880 to 2020 temperature increases around 0.8°C. But in your study you analyze the influence of 38°C temperature on pear compared to 25°C control condition. This is 13°C increment. So how 0.8°C temperature increase can influence pear development. My suggestion is to add more information about regional changes of temperature or regions with high temperature where the pear cultivation is very important.

Page 7, Line 242 - ...could cut genes (Supplementary Fig. 10). – rather induce siRNA production or induce RNA degradation but not cut genes.

Page 14, Line 488 - Supplementary Figure 4A – should be, according to the used style, Supplementary Fig. 4A

Reviewers' comments:

Reviewer #1 (Remarks to the Author):

I appreciate the corrections made by the authors. the additional explanations and data provided.

I also asked the author to limit their claims about the HILinc1. The authors have now provided a section in the discussion about the limitation of their methodology. I however would recommend them too also include the reference to the "transient nature of the method" in the abstract:

I would like to suggest amending the following sentence:

"Overexpressing either HILinc1 or PbHILT1 increases thermotolerance in pear, while silencing HILinc1 or PbHILT1 makes pear more heat sensitive. "

to

"Transient overexpression of either HILinc1 or PbHILT1 increases thermotolerance in pear, while transient silencing of HILinc1 or PbHILT1 makes pear plants more heat sensitive. "

Answer: Thanks for your suggestion. We have amended the sentence in the abstract at line 36-38 in red.

Please also correct the sentence:

line 442 "However, we also admitted that this transformation system was not stable and lasted for a short time with stable functions."

to

line 442 "However, we also admit that used transformation system was not stable and lasted for a short time."

Answer: Thanks for your suggestion. We have corrected the sentence at line 421-422 in red.

Reviewer #3 (Remarks to the Author):

Well done. I accept the revised article but still I have four notes, see below

Page 2, Line 40 - ... which was the target gene – rather target transcript.

Answer: Thanks for your suggestion. We have corrected the sentence at line 32 in red.

Page 2 Line 54 - ...around 0.8°C since 1880 (Janni et al., 2020)...- such difference between 1880 and 2020 could be explained by temperature measurement method. Answer: This is the reference we looked up about global temperature change. Sorry, we have no idea about what your question means.

Comment: You show that from 1880 to 2020 temperature increases around 0.8°C. But in your study you analyze the influence of 38°C temperature on pear compared to 25°C control condition. This is 13°C increment. So how 0.8°C temperature increase can influence pear development. My suggestion is to add more information about regional

changes of temperature or regions with high temperature where the pear cultivation is very important.

Answer: Thanks for your suggestion. We have added information about the changes of extremely high temperature in the major pear-producing areas in China at line 47-51.

In 2022, the average maximum temperature of major pear-producing areas in China (including Hebei, Anhui, Shandong, Henan, Shanxi and Zhejiang) was 38.8°C. The main materials in this study were tissue-cultured pear plantlets and 25°C is the culture temperature, which is suitable for pear growth. In order to study the heat stress response of pear, we chose 25°C treatment as the normal control and 38°C treatment as the heat stress group.

Page 7, Line 242 - ...could cut genes (Supplementary Fig. 10). – rather induce siRNA production or induce RNA degradation but not cut genes.

Answer: Thanks for your suggestion. We have corrected the sentence at line 227 in red.

Page 14, Line 488 - Supplementary Figure 4A – should be, according to the used style, Supplementary Fig. 4A

Answer: Thanks for your suggestion. We have corrected the statement at line 466 in red.